# Thermosensation in *Caenorhabditis elegans* is linked to ubiquitin-dependent protein turnover via insulin and calcineurin signalling

Alexandra Segref [1,2] ✉, Kavya L. Vakkayil [1,2], Tsimafei Padvitski[1,2], Qiaochu Li [1,2,6], Virginia Kroef[1,2,4,6], Jakob Lormann[1,2], Lioba Körner[1,2], Fabian Finger[1,2,5] & Thorsten Hoppe [1,2,3] ✉

Organismal physiology and survival are influenced by environmental conditions and linked to protein quality control. Proteome integrity is achieved by maintaining an intricate balance between protein folding and degradation. In *Caenorhabditis elegans*, acute heat stress determines cell non-autonomous regulation of chaperone levels. However, how the perception of environmental changes, including physiological temperature, affects protein degradation remains largely unexplored. Here, we show that loss-of-function of *dyf-1* in *Caenorhabditis elegans* associated with dysfunctional sensory neurons leads to defects in both temperature perception and thermal adaptation of the ubiquitin/proteasome system centered on thermosensory AFD neurons. Impaired perception of moderate temperature changes worsens ubiquitin-dependent proteolysis in intestinal cells. Brain-gut communication regulating protein turnover is mediated by upregulation of the insulin-like peptide INS-5 and inhibition of the calcineurin-regulated forkhead-box transcription factor DAF-16/FOXO. Our data indicate that perception of ambient temperature and its neuronal integration is important for the control of proteome integrity in complex organisms.

The ability to sense the external environment is vital for organisms to adapt and maintain homeostasis in response to both physiological and stressful changes[1]. The balance of protein homeostasis (proteostasis) including the stability of the entire proteome safeguards cellular integrity and organismal health[2]. Proteostasis is achieved by chaperone-dependent protein folding and selective degradation of proteins via the ubiquitin/proteasome-system (UPS) or the autophagy-lysosome pathway[3]. Impairment of these pathways can lead to the accumulation of damaged proteins which often form proteotoxic aggregates. The UPS is based on a cascade of three enzymes (E1-E3's) to attach the 76 amino acid polypeptide ubiquitin to proteins that are damaged or no longer required for cellular function and targets them for degradation by the 26 S proteasome. An E1 enzyme activates free ubiquitin in an ATP-dependent manner, an E2 enzyme attaches ubiquitin to an internal lysine residue of the substrate with the help of an E3 ligase that recognizes the specific substrate destined for

[1]Institute for Genetics, University of Cologne, 50674 Cologne, Germany. [2]Cologne Excellence Cluster on Cellular Stress Responses in Aging-Associated Diseases (CECAD), University of Cologne, 50931 Cologne, Germany. [3]Center for Molecular Medicine Cologne (CMMC), Faculty of Medicine and University Hospital of Cologne, 50931 Cologne, Germany. [4]Present address: Max Planck Institute for Biology of Ageing, 50931 Cologne, Germany. [5]Present address: Novo Nordisk Foundation Center for Basic Metabolic Research, University of Copenhagen, 2200 Copenhagen, Denmark. [6]These authors contributed equally: Qiaochu Li, Virginia Kroef. ✉e-mail: asegref@uni-koeln.de; thorsten.hoppe@uni-koeln.de

degradation by the 26 S proteasome[4–7]. Thereby, the UPS allows the cell to rapidly respond to changing demands by dynamic adaptation of protein stability.

It was shown that organismal homeostasis is governed by sensory neurons via cell non-autonomous coordination between different tissues. The neuronal system of *C. elegans* comprises 60 ciliated sensory neurons, which serve as antennas for the detection of external cues[8]. The ciliated wing neurons, called AWA, AWB, and AWC, and the amphid finger AFD neurons terminate in the sheath cells, while the others penetrate the cuticle to expose their cilia to the environment[8]. These sensory neurons transfer external information to interneurons for processing and coordination of organismal behaviour and physiology[9]. Ubiquitous neuronal expression of the transcription factor XBP-1 regulates endoplasmic reticulum (ER) and lipid homeostasis, thereby controlling cellular protein aggregation[10–12]. Furthermore, olfactory AWC neurons regulate intestinal lipid homeostasis through activation of the FOXO transcription factor DAF-16 via FLP-1 signalling[13] as well as lifespan[14,15]. Thermosensory AFD neurons regulate cellular heat shock response (HSR) upon acute heat shock, which triggers organism-wide adaptation of proteostasis[16,17]. However, it remains unclear how sensation of minor temperature changes controls protein degradation.

Here, we analyse the role of sensory neurons in ubiquitin-dependent protein turnover. Ciliary mutant worms exhibit defects in thermotaxis and UPS function which involves the role of AFD neurons. Perception of environmental temperature changes regulates organismal protein turnover by brain-to-gut communication via insulin and calcineurin signalling. Together, these data reveal that sensory ciliated neurons coordinate thermotaxis and ubiquitin-dependent protein turnover, which adjusts the dynamics of the organismal proteome in response to minor temperature changes.

## Results

### Ciliary neurons coordinate protein degradation by the UPS
To investigate the role of sensory perception in the regulation of protein turnover, we studied mutant worms lacking distal segments of sensory cilia[18,19]. Specifically, DYF-1 is a conserved ciliary protein that docks the motor kinesin-II OSM-3 to intraflagellar transport molecules during the formation of distal ciliary segments. Hence, *dyf-1* and *osm-3* mutant worms cannot build functional sensory cilia and thus cannot properly perceive external stimuli, reflected by their reduced chemotaxis to attractants and diminished osmotic avoidance behaviour[18–20]. To determine if ciliary mutant worms display protein turnover defects, we took advantage of an in vivo degradation assay based on the fluorescently labelled ubiquitin fusion degradation (UFD) model substrate UbV-GFP, which is targeted by the UPS[21,22]. The *dyf-1(mn335)* loss-of-function mutant displayed reduced uptake of the stain DiI (Fig. 1a), as expected due to defective sensory cilia[18,19], and increased levels of UbV-GFP in intestinal cells (Fig. 1a, b). Both cilia formation and UFD substrate degradation were rescued by transgenic expression of wild-type *dyf-1(+)*, indicating a correlation between environmental sensation and UFD substrate degradation (Fig. 1a, b). We also examined an *osm-3(n1540)* partial loss-of-function mutant and a *dyf-1(mn335); osm-3(n1540)* double mutant, which revealed non-synergistic UbV-GFP stabilisation, further supporting that this phenotype derives from the absence of functional cilia (Fig. 1c, Supplementary Fig. 1a, b). Since the *dyf-1(mn335)* and *dyf-1(mn335); osm-3(n1540)* showed a similar phenotype, we further focused on characterisation of the *dyf-1(mn335)* mutant.

To monitor UFD substrate degradation rates, we used cycloheximide to block protein translation. The rate of UFD substrate decay was reduced in *dyf-1(mn335)* compared with wild-type worms (Fig. 1d, e). As a control, we analysed a deletion mutant of *hecd-1*, which encodes the E3 ubiquitin ligase required for UFD substrate ubiquitylation; this mutation completely blocked its degradation (Fig. 1d, e). UFD

substrate expressed in the hypodermis was also stabilised in the *dyf-1(mn335)* mutant, suggesting that defects in ciliary neurons trigger a general response in different tissues (Supplementary Fig. 1c). The stabilised UFD substrate in the *dyf-1(mn335)* mutant was efficiently ubiquitylated (Fig. 1f); thus, substrate conjugation was not affected upon loss of functional sensory cilia. In addition, the stability of the ER substrate CPL-1$^{W32A,Y35A}$::YFP was unaffected compared with the *sel-1(e1948)* mutant[23], indicating that ER-associated protein degradation (ERAD) was unaffected in *dyf-1(mn335)* (Fig. 1g, h). The level of the *C. elegans* BiP homologue HSP-4 was also unaffected in *dyf-1(mn335)* (Supplementary Data 1, Supplementary Data 2), and general heat-shock proteins were not increased, indicating that there was no general stress response (Supplementary Fig. 1d, Supplementary Data 2). Taken together, these observations suggest that ciliated sensory neurons promote the degradation of ubiquitylated proteins without eliciting a general stress response.

### Neuronal signalling regulates UFD substrate degradation
DYF-1 function controls formation of the distal segments of cilia in sensory neurons and has been reported to be expressed in amphid, phasmid, and inner labial neurons in *C. elegans*[24]. We therefore addressed if specific loss of these neurons would recapitulate the UFD substrate degradation defect of the *dyf-1(mn335)* mutant. We expressed UbV-GFP in several mutants that affect the identity and function of amphid or phasmid neurons (Fig. 2a). *odr-7(tm4791)*, which affects AWA neuronal identity[25], displayed increased UbV-GFP levels in intestinal cells as compared with the other mutants (Fig. 2a–c). Thus, protein degradation in intestinal cells requires ciliated neurons in general, and AWA signalling in particular.

AWA are the primary sensory neurons presynaptic to the AFD neurons[26–28]. Interestingly, mutation of *gcy-8(oy44)*, a guanylyl cyclase that is specifically expressed in and required for AFD function[16,29,30], suppressed the UFD substrate stabilisation in *dyf-1(mn335)* (Fig. 2d, Supplementary Fig. 2a). ODR-7 determines the identity of AWA neurons by suppressing the expression of the G protein-coupled receptor STR-2, which is mainly asymmetrically expressed in AWC neurons[31–34]. Therefore, we investigated whether the *odr-7(tm4791)* mutation controls UPS turnover through *str-2* expression. Interestingly, *str-2(tm445); odr-7(tm4791)* mutant worms show a UPS defect similar to *dyf-1(mn335)*, and this effect is suppressed by the *gcy-8(oy44)* mutation (Fig. 2e). Thus, when AWA neuron identity is impaired, the lack of AWC$^{ON}$ identity in these neurons contributes to the UPS turnover defect upstream of AFD function. In addition, although defects in dense core vesicle secretion via UNC-31, the *C. elegans* ortholog of the calcium-dependent secretion activator (CADPS), or synaptic vesicle secretion via UNC-13[35] did not affect protein degradation in wild-type worms, the UFD substrate stabilisation was suppressed in *dyf-1(mn335); unc-31(e928)* and *dyf-1(mn335) unc-13(e51)* mutant worms (Supplementary Fig. 2b, c).

Because the *dyf-1(mn335)* mutant decreased protein turnover and, in addition, AFD function has been reported to affect protein misfolding[17], we investigated whether *dyf-1(mn335)* worms exhibit increased protein aggregation. We tested whether the *dyf-1(mn335)* mutant affects aggregation of an in vivo folding reporter used to model Huntington's disease, which consists of 44 poly-glutamine residues fused to YFP and is expressed in intestinal cells (Q44::YFP)[36]. Interestingly, *dyf-1(mn335)* mutant worms show significantly reduced numbers of polyQ aggregates compared with wild-type without affecting the Q44::YFP protein levels (Fig. 2f, g, Supplementary Fig. 2d, e), similar to the previously reported *gcy-8(oy44)* mutation[17]. These data suggest that in the *dyf-1(mn335)*, defective sensory ability via AWA activates AFD signalling, which in turn reduces UFD substrate degradation in the gut. However, the *dyf-1(mn335)* mutant exhibits reduced polyQ aggregation, similar to the previously reported *gcy-8(oy44)* mutant[17]. To determine whether the aberrant neuron

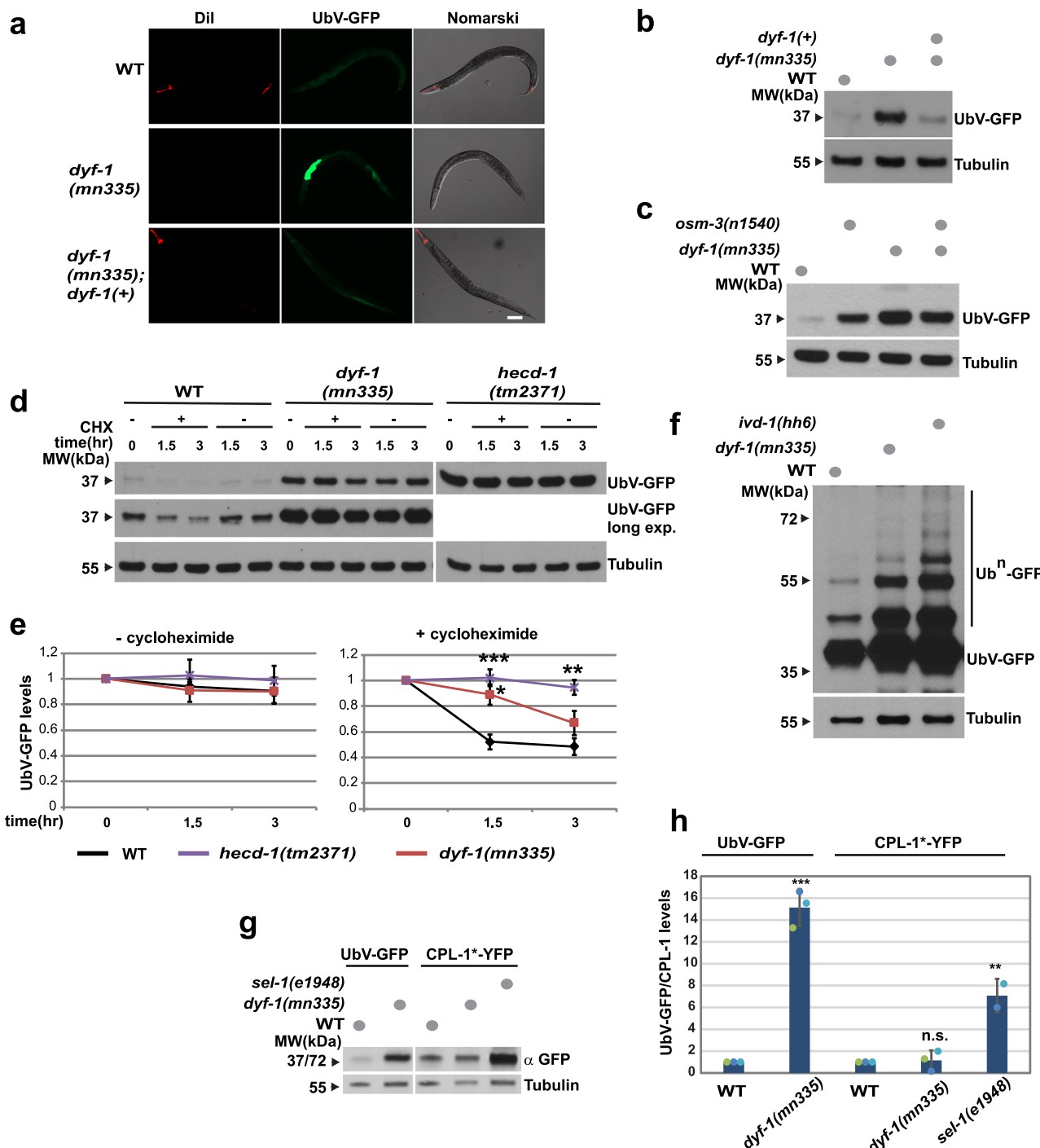

**Fig. 1 | Defective sensory cilia mutants show reduced UPS-dependent protein degradation. a** Fluorescence and Nomarski images of worms expressing the UFD substrate in WT, *dyf-1(mn335); dyf-1(+)* and the direct sibling lacking the *dyf-1* array *(dyf-1(mn335))*. DiI: ciliated amphid and phasmid neurons. Scale bar: 100 μm. **b** Protein lysates of worms displayed in (**a**) were analysed by Western blotting against GFP and tubulin. Dots: presence of indicated mutation. **a**, **b** $n = 3$ biologically independent experiments. **c** Protein lysates of indicated WT and mutant worms. $n = 5$ biologically independent experiments. **d** Protein lysates of wild-type, *dyf-1(mn335)* or *hecd-1(tm2371)* worms after ethanol (−) or 0.5 mg/mL cycloheximide in ethanol treatment for the indicated time points. Worms were analysed as in (**b**). long exp. = long exposure of the blot. **e** Quantification of 4 biologically independent experiments from (**d**), average ± standard deviation, one-way ANOVA with Bonferroni's multiple comparisons test, *hecd-1(tm2371)* 1.5 h $p = 0.0004$, *hecd-1(tm2371)* 3 h $p = 0.0012$, *dyf-1(mn335)* 1.5 h $p = 0.0166$, *dyf-1(mn335)* 3 h $p > 0.9999$, ratios of mutants versus ratios of wild-type with cycloheximide. **f** WT and indicated

mutant worms expressing the UFD substrate were analysed as in (**b**), but the top part of the blot exposed for an extended time to detect polyubiquitylated UbV-GFP, *ivd-1(hh6)*: positive control to detect polyubiquitylated UbV-GFP[56], $n = 1$. **g** WT or indicated mutant worms expressing UbV-GFP or the procathepsin L mutant CPL-1[W32A,Y35A]::YFP (CPL-1*-YFP), a marker for ER-associated protein degradation, are analysed by Western blotting against GFP and tubulin. **h** Quantification of the experiment displayed in (**g**) Column graphs: mean ± standard deviation, scatter plots: individual experiments, UbV-GFP: *dyf-1(mn335)* versus WT $p = 0.0001$, 2-tailed unpaired Student's *t*- test, CPL-1*: *dyf-1(mn335)* versus WT $p = 0.97$, *sel-1(e1948)* versus WT $p = 0.0013$, One-way ANOVA followed by Dunnett's multiple comparisons test, $n = 3$ (WT and *dyf-1(mn335)*) and $n = 2$ (*sel-1(e1948)*) biologically independent experiments. **e**, **h** ****$p \leq 0.0001$, ***$p \leq 0.001$, **$p \leq 0.01$, *$p \leq 0.05$, n.s. $p > 0.05$. See also Supplementary Fig. 1. Source Data are provided as a Source Data file.

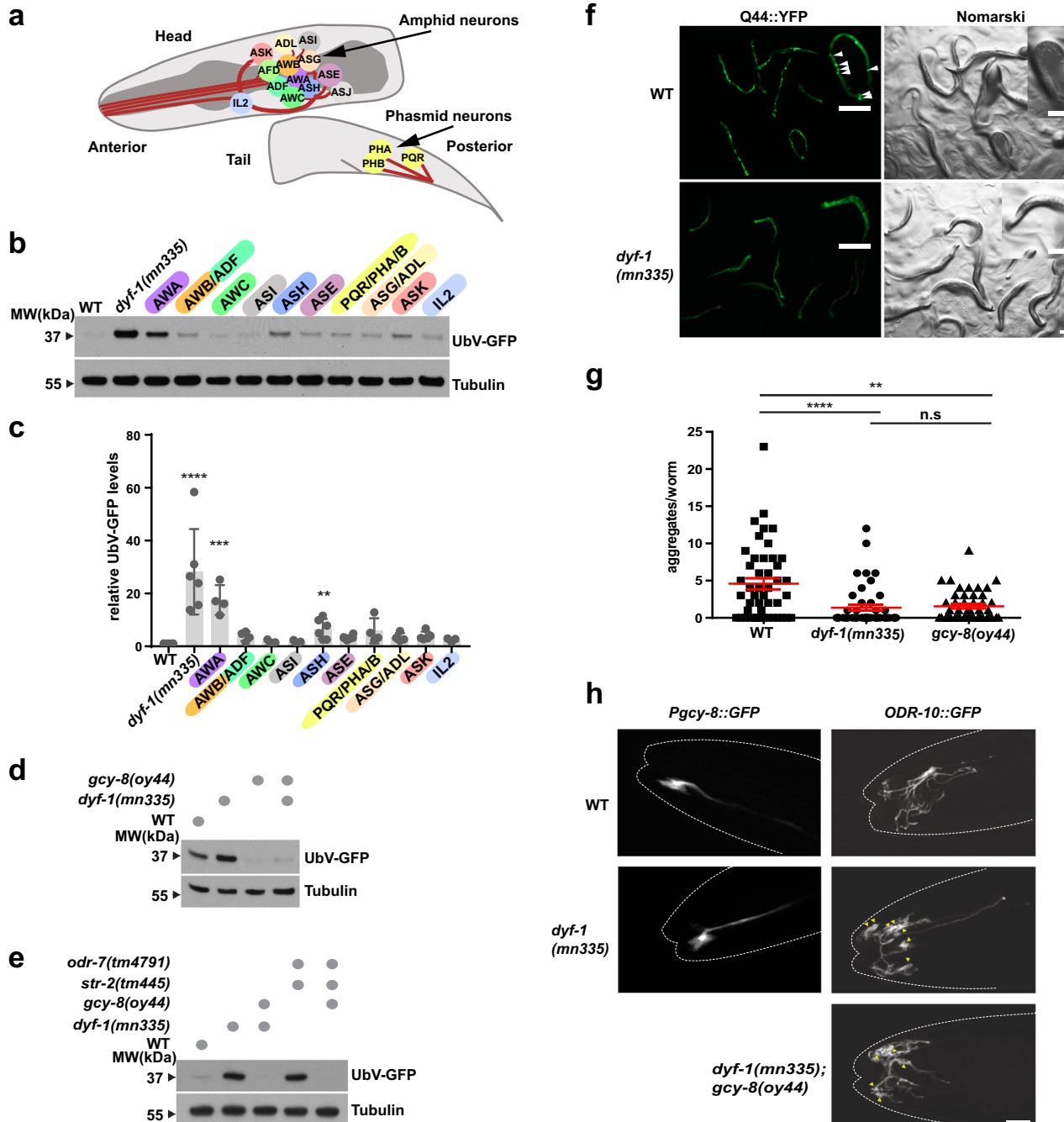

**Fig. 2 | Neuronal signalling controls ubiquitin-dependent turnover and PolyQ aggregation. a** Schematic illustration of analysed *C. elegans* ciliated head and tail neurons. **b** WT, *dyf-1(mn335)*, or worms harbouring mutations affecting the functionality of indicated neurons and expressing UbV-GFP were analysed by Western blotting against GFP and tubulin. Mutants: AWA *odr-7(tm4791)*, AWB/ADF *lim4(ky403)*, AWC *ceh-36(ks86)*, ASI *unc-3(e151)*, ASH *unc-42(e270)*, ASE *che-1(ot75)*, PQR, PHA, PHB *ceh-14(ch3)*, ASG/ADL *lin-11(n389)*, ASK *ttx-3(ot22)*, and IL2 *unc-86(n847)*. **c** Quantification of worms from (**b**), *n* = 6 (WT, *dyf-1*, ASH, ASE), 5 (ASI, ASK), 4 (AWA, AWB/ADF, PQR, PHA, PHB, ASG/ADL) and 3 (IL2, AWC) biologically independent experiments. Column graphs: mean ± standard deviation, scatter plots: individual experiments, *dyf-1(mn335)* p = <0.0001, *odr-7(tm4791)* p = 0.001 (including *dyf-1(mn335)* for maximum changes), *lim4(ky403)* p = 0.5413, *ceh-36(ks86)* p = 0.9996, *unc-3(e151)* p = 0.9996, *unc-42(e270)* p = 0.0067, *che-1(ot75)* p = 0.6447, *ceh-14(ch3)* p = 0.0528, *lin-11(n389)* p = 0.708, *ttx-3(ot22)* p = 0.4099, *unc-86(n847)* p = 0.9867 (excluding *dyf-1(mn335)* from comparison to detect significant changes

compared with WT), One-way ANOVA with Dunnett's multiple comparisons test. **d**, **e** Indicated mutant worms analysed as in (**b**), *n* = 2 biologically independent experiments (**d**, **e**). **f** Fluorescence and Nomarski images of day 1 WT or *dyf-1(mn335)* worms expressing intestinal Q44::YFP. Scale bars: 200 µm. **g** Quantification of Q44::YFP aggregates per day 1 worm. Red: mean ± s.e.m. WT versus *dyf-1(mn335)* p < 0.0001, *gcy-8 (oy44)* p = 0.0072, *dyf-1(mn335)* versus *gcy-8(oy44)* p = 0.4192, Kruskal–Wallis one-way ANOVA with Dunn's multiple comparisons test, n = 2 biologically independent experiments (**c**, **g**) ****p ≤ 0.0001, ***p ≤ 0.001, **p ≤ 0.01, n.s. p > 0.05. **h** Confocal images of the finger-shaped amphid AFD (*Pgcy-8::GFP*, villi and cilium) and branched AWA (ODR-10::GFP) cilia in WT and indicated mutants. Anterior: left. Arrowheads: accumulation of ODR-10::GFP in the cilia. Typical results of (*Pgcy-8::GFP*: 27 (WT), 30 (*dyf-1(mn335)*); *odr-10::GFP*: 45 (WT), 29 *dyf-1(mn335)*, 37 *dyf-1(mn335)*; *gcy −8(oy44)* worms, *n* = 2 biologically independent experiments. Scale bar 5 µm. See also Supplementary Fig. 2f, Source data are provided as a Source Data file.

morphology contributes to the UPS defects, we analysed the morphology of AFD and AWA neurons in the *dyf-1(mn335)* mutant by expressing *Pgcy-8::GFP* and ODR-10::GFP, respectively[37]. We observed no structural changes in the microvilli and cilia structures of AFD neurons in the *dyf-1(mn335)* mutant compared with wild-type (Fig. 2h). However, AWA neurons exhibit cilia structures showing accumulation of ODR-10::GFP in the *dyf-1(mn335)* mutant compared with wild-type, which was not rescued by *gcy-8(oy44)* mutation (Fig. 2h, Supplementary Fig. 2f). Thus, impaired UFD turnover correlates with structural abnormalities in AWA cilia and AFD neuron activity.

### Endogenous substrate degradation regulated by ciliated neurons

To determine if endogenous proteins are also regulated in the absence of functional sensory cilia, we performed mass spectrometric analysis of wild-type, *dyf-1(mn335)*, *dyf-1(mn335); unc-31(e928)* and *dyf-1(mn335) unc-13(e51)* worms (Fig. 3a, b). Several proteins were significantly increased in the *dyf-1(mn335)* mutant compared with wild-type (Fig. 3a, Supplementary Data 1) Among those proteins several were decreased upon inhibition of neuronal signalling in *dyf-1(mn335); unc-31(e928)* or *dyf-1(mn335) unc-13(e51)* worms (Fig. 3b, Supplementary Fig. 3a, b). To verify the mass spectrometric analysis, we followed protein turnover of selected proteins. Amine N-methyltransferase ANMT-2 and the putative hydroxypyruvate isomerase C05D11.5 proteins were increased in *dyf-1(mn335)* worms (Fig. 3b), which was not linked to mRNA levels, as detected by microarray analysis of wild-type and *dyf-1(mn335)* (Supplementary Fig. 3c, Supplementary Data 2). These data are consistent with defective post-translational regulation of ANMT-2 and C05D11.5 specifically in *dyf-1(mn335)* mutant worms, which also stabilise the UFD substrate UbV-GFP.

We endogenously tagged both proteins with GFP, and detected increased levels of both ANMT-2-GFP and GFP-C05D11.5 in *dyf-1(mn335)* (Supplementary Fig. 3d–g). RNAi-mediated depletion of the proteasomal subunit RPN-8 also increased the levels of both proteins in wild-type worms but not in *dyf-1(mn335)* (Fig. 3c–f), indicating that ANMT-2-GFP and GFP-C05D11.5 are not degraded by the 26 S proteasome upon sensory defects. ANMT-2 and C05D11.5 levels were reduced in the *dyf-1(mn335); unc-31(e928)* mutant compared with *dyf-1(mn335)*, consistent with a role for neuronal signalling in UFD substrate stabilisation in the *dyf-1(mn335)* mutant (Supplementary Fig. 4a–d). GCY-8 was required for GFP-C05D11.5 stabilisation in the *dyf-1(mn335)* mutant (Supplementary Fig. 4c, d). Thus, we observe an AFD-dependent accumulation of endogenous proteins in the absence of functional sensory cilia.

### FLP-3 and INS-5 peptides inhibit the UPS in the *dyf-1(mn335)* mutant

Our results suggest that defective sensory cilia trigger inhibition of ubiquitin-dependent protein degradation in intestinal cells by amplifying a neuronal signal. Gene enrichment analysis of microarray data revealed that genes involved in mitochondrial function and protein translation were decreased, whereas genes involved in neuronal signalling pathways were predominantly increased in the *dyf-1(mn335)* mutant compared with wild-type (Fig. 4a, Supplementary Data 2). Transcripts encoding several potential neuropeptides were elevated in *dyf-1(mn335)* compared with wild-type (Fig. 4b, Supplementary Data 2), which we verified by qRT-PCR (Supplementary Fig. 5a, b). To test if these peptides mediate the neuronal signalling and resulting protein degradation defects in *dyf-1(mn335)* worms, we combined neuron-sensitised RNAi and genetic analyses. Among all candidates tested (Supplementary Table 1), depletion of the neuropeptide FLP-3 and the insulin-like peptide INS-5 restored UPS-dependent degradation in *dyf-1(mn335)* worms, as verified by mutational analysis (Fig. 4c, d, Supplementary Fig. 5c, d, and Supplementary Table 1). We infer that FLP-3 and INS-5 negatively regulate

UPS-dependent degradation in the intestine in the *dyf-1(mn335)* mutant.

FLP-3 belongs to a family of 31 FMRF amide-related peptides (FLP)[38], and is expressed in neurons[39,40], suggesting that it is aberrantly upregulated in neurons of the *dyf-1(mn335)* mutant. On the other hand, INS-5, which is a member of a family of 40 insulin-like peptides discovered in *C. elegans*[41–43], is expressed in several tissues, including neurons and the intestine[44,45]. Because INS-5 is expressed in different tissues with greater expression changes compared with FLP-3 (Supplementary Data 2), we wanted to test whether neuronal and/or intestinal INS-5 contributes to the UFD substrate degradation defects of *dyf-1(mn335)*. We re-expressed *ins-5* from different cell-type specific promotors in *dyf-1(mn335); ins-5(tm2560)* double mutant worms. We verified that *ins-5* expressed from its own promoter in *dyf-1(mn335); ins-5(tm2560)* worms phenocopied the accumulation of the UFD substrate observed in the *dyf-1(mn335)* worms (Fig. 4e, Supplementary Fig. 5e). Although *ins-5* expressed from the neuronal promoter *unc-119* did not lead to UFD substrate accumulation, *ins-5* expressed under the intestinal *ges-1* promoter stabilised the UFD substrate in the *dyf-1(mn335); ins-5(tm2560)* worms comparable to *dyf-1(mn335)* alone (Fig. 4f, Supplementary Fig. 5f–h). We confirmed that *unc-119-* and *ges-1* promoters induced similar levels of *ins-5* expression (Supplementary Fig. 5f, g). Together, these findings suggest that elevated INS-5 expression in the intestine inhibits UPS-dependent degradation in the *dyf-1(mn335)* mutant lacking sensory capability.

### Calcineurin signalling inhibits the UPS in *dyf-1(mn335)*

Given that defects in UNC-31 and thus calcium-dependent vesicle secretion suppressed the stabilisation of the UFD substrate in the *dyf-1(mn335)* mutant, we further examined the calcium signalling pathway. Transcript levels of several members of this pathway were elevated in *dyf-1(mn335)* compared with wild-type, including the calcineurin A ortholog TAX-6 (CaN) (Fig. 5a). Intriguingly, reducing the expression of TAX-6 suppressed the protein degradation defect but not the cilia defect of *dyf-1(mn335)* (Fig. 5b, Supplementary Fig. 6a, and Supplementary Table 1). TAX-6 has been reported to negatively regulate the function of the FOXO transcription factor DAF-16[46]. Given that DAF-16 is known to influence proteostasis[47], we hypothesized that reduced TAX-6 levels suppress the inhibited UPS in the *dyf-1(mn335)* mutant by activating DAF-16. Indeed, mutation of *daf-16* in the *dyf-1(mn335); tax-6(p675)* double mutant phenocopied the UFD substrate accumulation of the *dyf-1(mn335)* single mutant (Fig. 5c). In contrast, *daf-16(mu86)* has no effect on UFD degradation in the wild-type or the *tax-6(p675)* mutant, suggesting that TAX-6 and DAF-16 do not regulate UFD degradation under these conditions (Fig. 5c).

To determine if the crosstalk between TAX-6 and DAF-16 occurs in neuronal or intestinal cells, we re-expressed *daf-16* from the *unc-119* or *ges-1* promoters in the *daf-16(mu86) dyf-1(mn335); tax-6(p675)* triple mutant[48]. Intestinal, but not neuronal, expression of *daf-16* fully suppressed inhibition of the UPS in the triple mutant (Fig. 5d). These data suggest that increased TAX-6 in the intestine of the *dyf-1(mn335)* mutant inhibits DAF-16 to diminish UPS-dependent protein degradation. Mutation of *gcy-8* abolished UFD substrate accumulation in the *dyf-1(mn335)* background (Fig. 2d, e). Similarly, loss of *gcy-8* abolished the accumulation of the UFD substrate in *tax-6(RNAi) dyf-1(mn335) daf-16(mu86)* worms (Fig. 5e). Thus, inhibiting AFD function in *dyf-1(mn335)* eliminates the TAX-6/DAF-16-mediated inhibition of UPS activity (Fig. 5f). UFD stabilisation was suppressed by *ins-5(tm2560)* even in the *dyf-1(mn335) daf-16(mu86)* double mutant background, indicating that the INS-5-dependent inhibition of UPS activity does not require DAF-16 (Fig. 5g, h).

To verify that loss of GCY-8 and TAX-6 do not reduce substrate expression in the *dyf-1(mn335)* mutant, we depleted the E3 ubiquitin ligase HECD-1. UbV-GFP levels were stabilised upon *hecd-1(RNAi)* in *gcy-8(oy44)* and *tax-6(p675)* mutants, confirming that these mutants restore

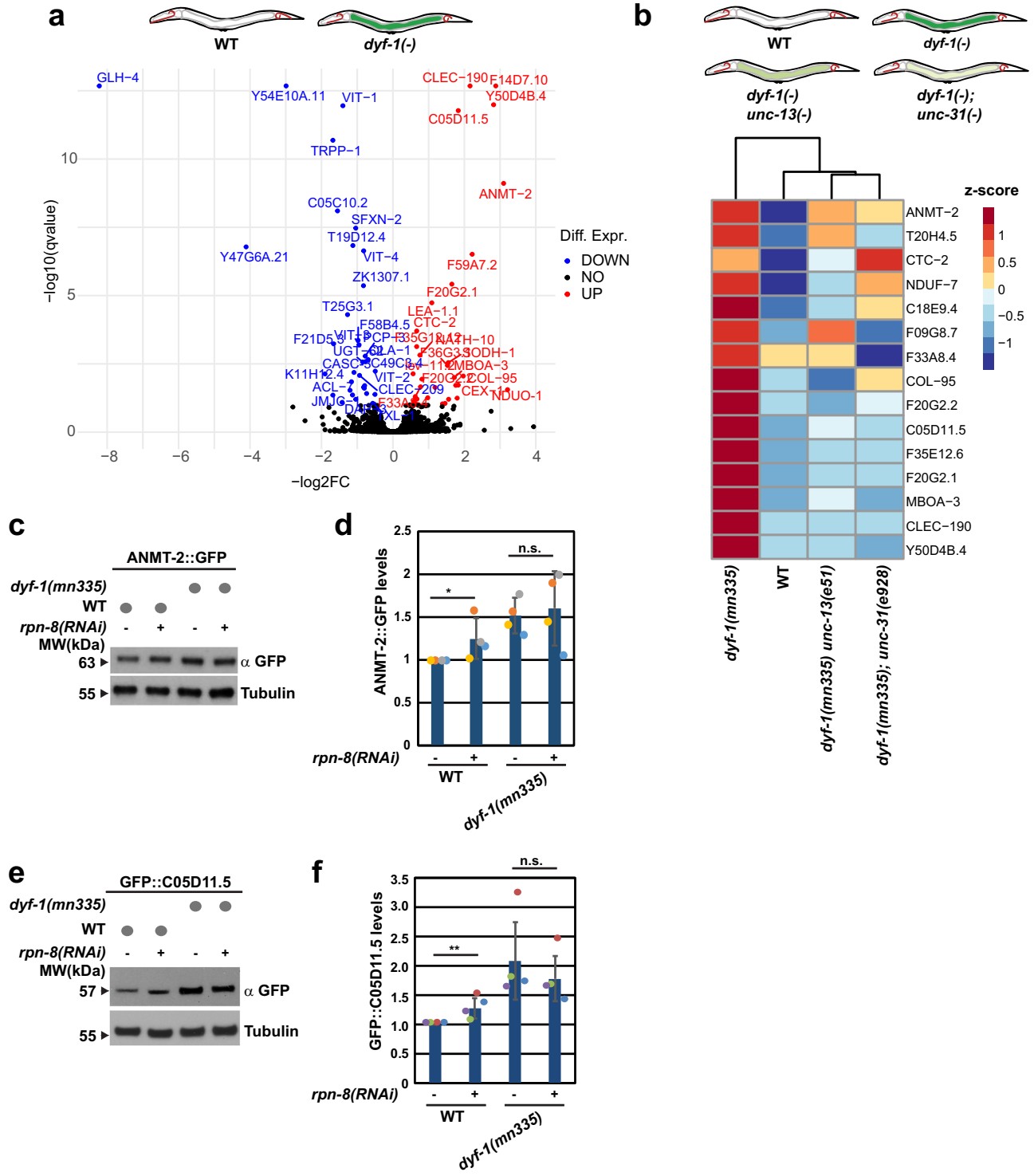

UPS activity in the *dyf-1(mn335)* mutant background (Supplementary Fig. 6b). Together, our data indicate that AFD neurons are activated in the *dyf-1(mn335)* mutant, triggering TAX-6 and INS-5-dependent inhibition of the UPS in the intestine. Because *C. elegans* sense and integrate thermal information in AFD neurons[16,49,50], we investigated whether the defect in protein turnover in the *dyf-1(mn335)* mutant is associated with a defect in temperature sensation.

### *dyf-1(mn335)* displays aberrant thermosensation

Worms memorise their cultivation temperature in the presence of food, and migrate towards this temperature if they are placed on a thermal gradient in the absence of food[51]. This response is called thermotaxis. To examine thermotaxis, we cultured worms at 22 °C and placed them at 25 °C within a 17–30 °C gradient (Fig. 6a–c). As expected, wild-type worms migrated towards 22 °C, whereas thermosensory *ttx-3(ks5)* mutant worms moved towards the region below 18 °C while *gcy-8(oy44)* mutant worms distributed in all temperature areas (Fig. 6c, Supplementary Table 2). A significant fraction of *dyf-1(mn335)* worms did not display thermotaxis and remained in regions above 24 °C, which was partially suppressed in the *dyf-1(mn335); gcy-8(oy44)* double mutant (Fig. 6c). When worms were cultivated at 15 °C and subsequently placed at 19 °C, wild-type and *ttx-3(ks5)* worms, and

**Fig. 3 | Mass spectrometry identifies endogenous substrates. a** Volcano plot shows significantly (*q*-value <0.1) increased (red) or decreased (blue) proteins in the *dyf-1(mn335)* mutant compared with WT derived by differential expression analysis of proteomic data. 3922 proteins were analysed in total, *n* = 3 biologically independent experiments. **b** Heat map of normalised protein intensity values (z-scores) derived by proteomic analysis of the indicated mutant strains. Proteins significantly (*q*-value <0.1, FDR corrected) increased in *dyf-1(mn335)* compared with WT and increased (*p*-value <0.05) compared with *dyf-1(mn335) unc-13(e51)* or *dyf-1(mn335); unc-31(e928)* double mutants are displayed, 2-sided, unpaired moderated *t*-test as calculated by limma package, *n* = 3 biologically independent experiments. Depicted schematic worms display the genotypes used and the expected levels of UbV-GFP substrate stabilisation. **c** Worms expressing the indicated mutations and ANMT-2::GFP were grown on *control* (−) or *rpn-8(RNAi)* (+) plates and analysed by Western

blotting against GFP and tubulin, *n* = 4 biologically independent experiments. **d** Quantification of the experiments displayed in (**c**), *n* = 4 biologically independent experiments. WT versus *rpn-8(RNAi)* *p* = 0.04256; *dyf-1(mn335)* vs *dyf-1(mn335) rpn-8(RNAi)* *p* = 0.36806. 1-tailed unpaired Student's *t*- test. **e** Worms expressing the indicated mutations and GFP::C05D11.5 were grown on *control* (−) or *rpn-8(RNAi)* (+) plates and analysed by Western blotting against GFP and tubulin, *n* = 4 biologically independent experiments. **f** Quantification of the experiments displayed in (**e**), *n* = 4 biologically independent experiments. WT versus *rpn-8(RNAi)* *p* = 0.0145; *dyf-1(mn335)* vs *dyf-1(mn335) rpn-8(RNAi)* *p* = 0.2621. 1-tailed unpaired Student's *t*- test. **d**, **f** Column graphs: mean ± standard deviation, scatter plots, individual experiments, same colour: same experiment, *\*p* ≤ 0.05, *\*\*p* ≤ 0.01, n.s. *p* > 0.05. See also Supplementary Fig. 3 and 4 and Supplementary Data 1, Source data are provided as a Source Data file.

to a lower extent *gcy-8(oy44)* worms, moved towards the region below 18 °C (Fig. 6d, e, Supplementary Table 3). Again, a large proportion of *dyf-1(mn335)* worms remained in the area of 18–23.5 °C, which was different in *dyf-1(mn335); gcy-8(oy44)* double mutants (Fig. 6d, e, Supplementary Table 3, Supplementary Fig. 7a–d). Similarly, worms with defects in AWA identity showed defective negative thermotaxis behaviour affected by the *gcy-8(oy44)* mutation (Fig. 6f, Supplementary Table 4). Thus, ciliary mutant worms display aberrant thermotaxis, which is partially suppressed by attenuating AFD-dependent signalling via GCY-8.

### Sensory neurons co-regulate thermal adaptation and protein degradation

We found that *dyf-1(mn335), gcy-8(oy44), and dyf-1(mn335); gcy-8(oy44)* mutants display distinct thermosensory defects. To investigate the relationship between thermal adaptation and protein turnover, we examined UFD substrate degradation at 15 °C, 20 °C, and 22 °C. In wild-type worms, UbV-GFP levels decreased with increasing temperature, due to elevated protein turnover (Fig. 7a, b, Supplementary Fig. 7e, f). Unexpectedly, UbV-GFP levels accumulated with increasing temperatures in *dyf-1(mn335)* worms (Fig. 7a, b), whereas UbV-GFP was efficiently degraded at all temperatures in *gcy-8(oy44)* and *dyf-1(mn335); gcy-8(oy44)* worms (Fig. 7c). Thus, *dyf-1(mn335)* worms, which do not properly sense temperature, display AFD-dependent inhibition of UFD substrate turnover despite increasing temperatures. Loss of AFD signalling, which leads to distinct defects in thermosensation, drives efficient UFD substrate turnover at all temperatures in the *dyf-1(mn335)* mutant and at low temperatures in wild-type worms (Fig. 7c). Next, we investigated whether the defects in thermosensation and protein turnover affected the overall fitness of the mutant worms by measuring their lifespan. We found that *dyf-1(mn335)* worms had a prolonged lifespan compared to wild-type, while *gcy-8(oy44)* mutant worms were short-lived, recapitulating genetic mutation or laser ablation experiments of AFD neurons[52–54]. The *gcy-8(oy44)* mutation suppressed the lifespan extension of the *dyf-1(mn335)* mutant (Supplementary Fig. 7g, Supplementary Table 5), similar to the dominant effect of this mutation on protein turnover. We conclude that sensory neurons communicate through AFD neurons to integrate ambient temperature with protein turnover and longevity. These data lead to a model that AFD signalling inhibits UFD substrate turnover at relatively low temperatures, which is counteracted by sensory neurons at higher temperatures (Fig. 8). We conclude that sensory neurons communicate via AFD to integrate ambient temperature changes with protein turnover.

## Discussion

Here, we provide evidence that thermosensation controls ubiquitin-dependent protein degradation via neuronal signalling. Worms deficient for environmental contact[18–20] display reduced ubiquitin-dependent protein degradation due to increased insulin and

calcineurin signalling. Knock-out of *ins-5* or *tax-6*, components of the insulin and calcineurin pathway respectively, do not further improve the UFD degradation observed in wild-type worms at 22 °C. However, *ins-5* and *tax-6*, as well as AFD neuron activity, are required for reduced UPS activity in *dyf-1(mn335)* worms lacking the distal segments of sensory cilia (Fig. 8a). We propose that lack of sensory perception via at least AWA in *dyf-1(mn335)* worms aberrantly activates AFD neuronal signalling to reduce UPS activity in the intestine and other tissues. Thus, environmental sensation via ciliated sensory neurons remodels multiple signalling pathways to regulate protein turnover.

INS-5 belongs to a family of 40 insulin-like peptides[55]. In the canonical insulin signalling pathway, insulin binding to the insulin receptor DAF-2 promotes a phosphorylation cascade regulating the activity of the FOXO transcription factor DAF-16, which controls diverse pathways including fat storage, body size, dauer formation, and lifespan[55,56]. We describe an additional role for INS-5 in negatively regulating ubiquitin-dependent protein degradation in *dyf-1(mn335)* worms, independently of DAF-16. We propose that it acts in a non-canonical insulin signalling pathway to inhibit protein degradation. So far DAF-2 is the most well studied insulin receptor in *C. elegans*, however recently additional receptors have been predicted by bioinformatic analysis and it is possible that INS-5 acts on a receptor other than DAF-2 to control UPS activity[57].

In the calcineurin pathway, the kinase CAMKII phosphorylates and activates DAF-16 whereas the phosphatase TAX-6 dephosphorylates and inactivates DAF-16 to mediate lifespan[46,58]. We show that *dyf-1(mn335)* worms with dysfunctional sensory cilia upregulate TAX-6, inhibiting DAF-16 and reducing UPS activity. Canonically insulin-signalled DAF-16 was previously shown to affect proteostasis on a global scale by affecting multiple pathways such a protein translation, detoxification, or DNA repair[47] but the targets associated to calcineurin-signalled DAF-16 are not well studied to date. DAF-16 activity is likely affecting the UPS in multiple ways and therefore TAX-6 depletion may entirely suppress UPS degradation defects in *dyf-1(mn335)*. We propose that the *dyf-1(mn335)* mutant perceives the loss of sensation as a mild stress which acts on the UPS through INS-5. However, loss of *tax-6* activity enhances the stress level, thereby activating DAF-16 to promote global proteostasis networks.

We show that slight changes in ambient temperature affect proteasomal degradation via neuronal signalling. Wild-type worms show increased levels of the UFD substrate at 15 °C compared to 22 °C, similar to previous observations of reduced UFD substrate levels in worms exposed to noxious or elevated heat temperature[22,59], but this effect is lost in the *gcy-8(oy44)* mutant that shows reduced thermotaxis and the absence of thermoreceptor currents[30] indicative of reduced AFD function. These data suggest that AFD neurons inhibit the degradation of proteasomal substrates at low temperature, and further that this AFD neuronal activity is suppressed at higher temperature. Sensation of elevated temperature may increase protein degradation through enhancing the

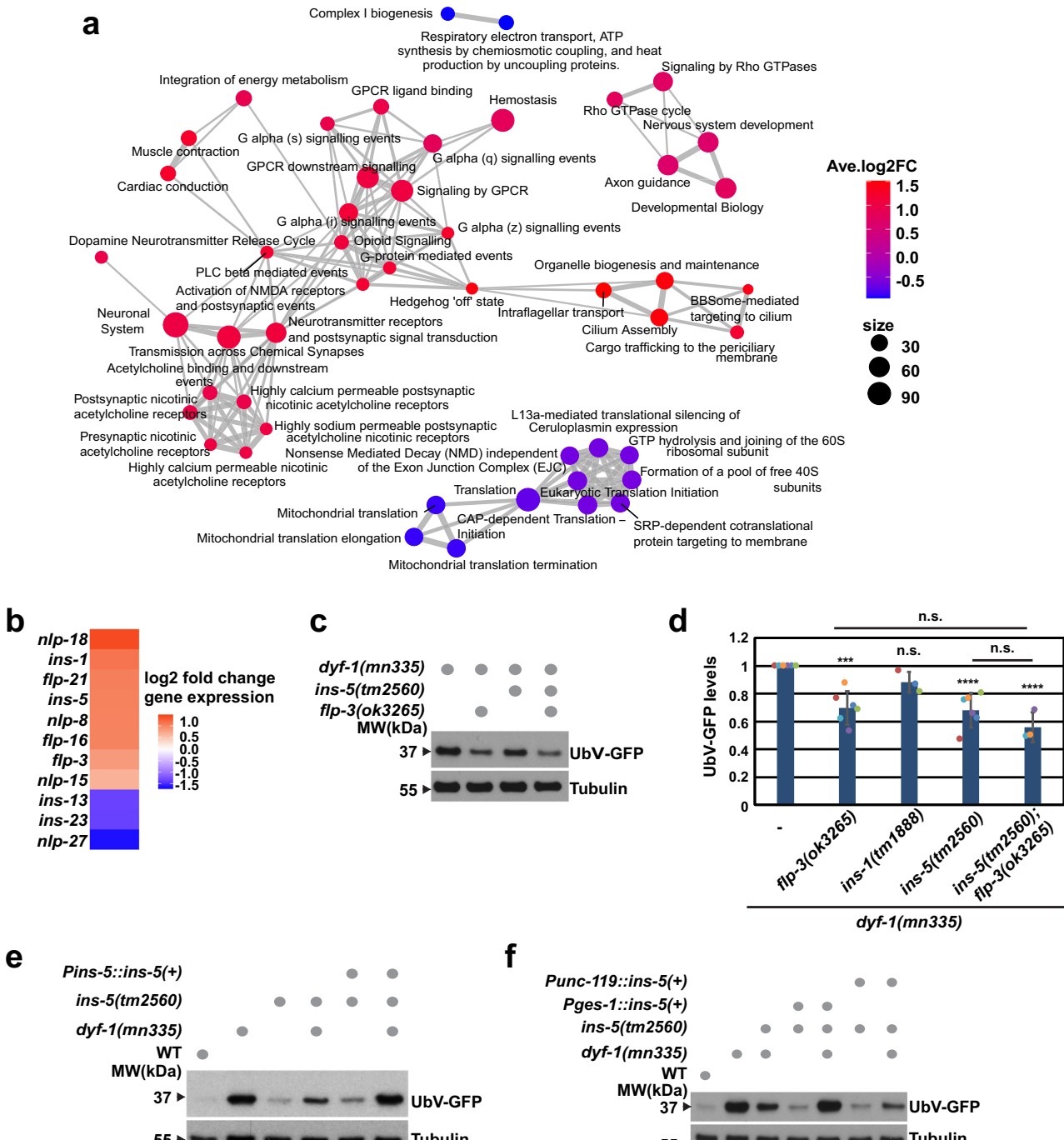

**Fig. 4 | FLP-3 and INS-5 signalling peptides control UPS-dependent protein turnover. a** Pathways (Reactome) significantly (*q*-value < 0.01) affected in *dyf-1(mn335)* compared with WT identified by GSEA (Methods), based on microarray data. Each pathway is represented by a bubble, size of a bubble: size of the pathway (number of its gene members affected). Thickness of edges reflects proportion of genes shared between 2 pathways. Colour of nodes reflects average change of the pathway gene members, red: the genes are upregulated, blue: genes are down-regulated. **b** Heatmap of differentially expressed neuropeptides (*q*-value < 0.01) in *dyf-1(mn335)* compared with WT based on microarray analysis. Colour of cells: log2 fold change of gene expression in *dyf-1(mn335)* compared with WT, *n* = 4 biologically independent experiments. **c** Worms expressing the UFD substrate and the mutations indicated were grown and analysed as in (Fig. 2b). **d** Quantification of UbV-GFP levels of worms from (**c**) and additionally *dyf-1(mn335); ins-1(tm1888)*. 6

*(dyf-1(mn335), dyf-1(mn335); flp-3(ok3265), dyf-1(mn335); ins-5(tm2560))* and 3 *(dyf-1(mn335); ins-1(tm1888), dyf-1(mn335); ins-5(tm2560); flp-3(ok3265))* biologically independent experiments. *dyf-1(mn335); flp-3(ok3265) p* = 0.0002, *dyf-1(mn335); ins-1(tm1888) p* = 0.312, *dyf-1(mn335); ins-5(tm2560) p* < 0.0001, *dyf-1(mn335); ins-5(tm2560); flp-3(ok3265) p* < 0.0001 compared with *dyf-1(mn335), dyf-1(mn335); flp-3(ok3265)* versus *dyf-1(mn335); ins-5(tm2560); flp-3(ok3265) p* = 0.1994, *dyf-1(mn335); ins-5(tm2560)* versus *dyf-1(mn335); ins-5(tm2560); flp-3(ok3265) p* = 0.2797. Column graph: mean ± standard deviation, scatter plots: individual experiments, same colour: same experiment ****p* ≤ 0.0001, ***p* ≤ 0.01, n.s. *p* > 0.05. One-way ANOVA followed by Dunnett's multiple comparisons test. **e, f** Indicated mutant worms expressing the UFD substrate were analysed as in (**c**). Typical result of *n* = 3 (**e**) and 4 (**f**) biologically independent experiments. Additional quantifications see Supplementary Fig. 5, Source data are provided as a Source Data file.

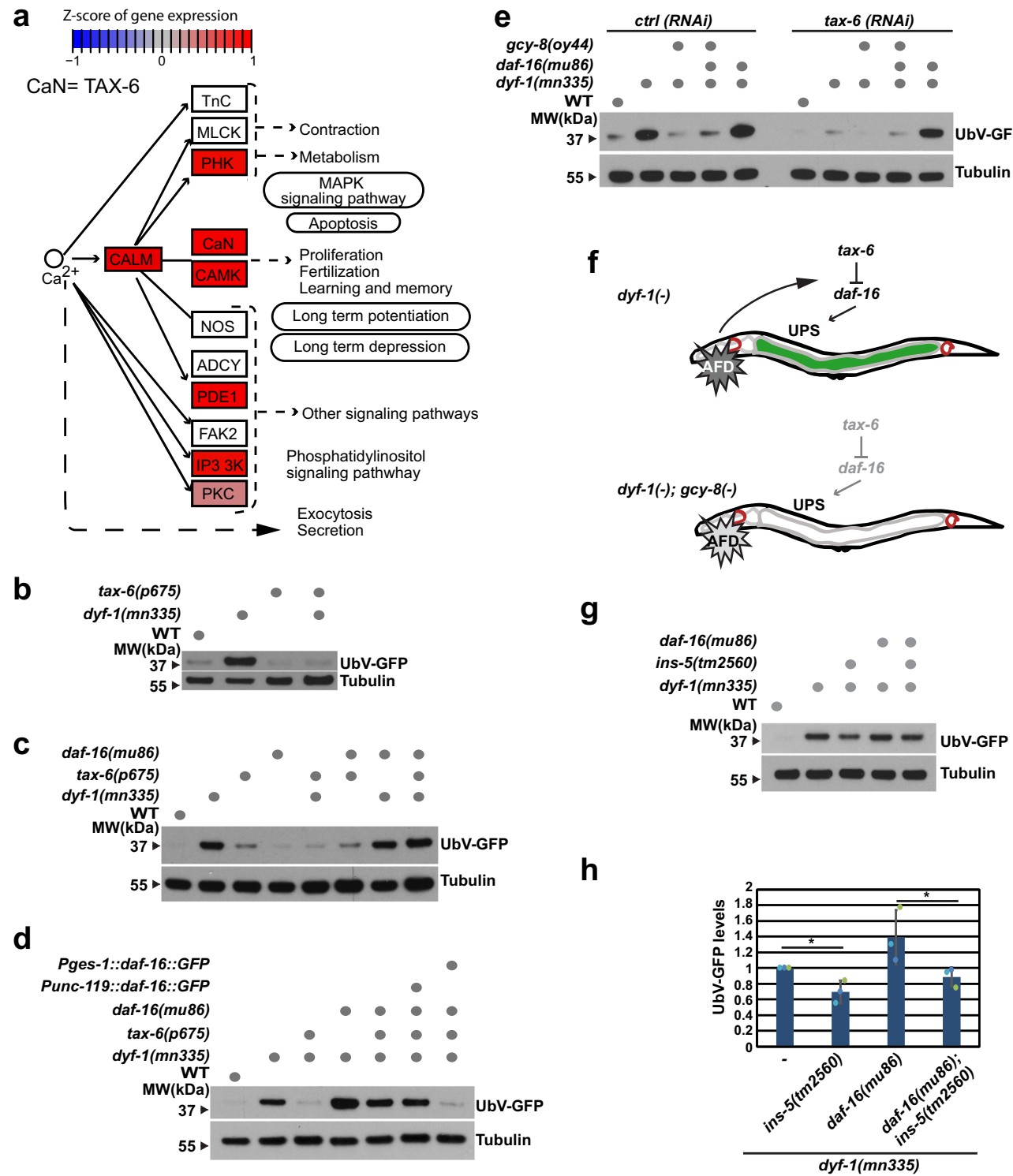

kinetics of proteasomal activity. This could be achieved through release of the proteasome from negative regulation[60,61]. Alternatively, the amount of proteasomal targets may result in an overload of the proteasome at low temperature slowing down overall degradation rate. In contrast, the *dyf-1(mn335)* mutant lacking proper sensory capability shows reduced protein degradation at elevated temperature. This response depends on AFD function, which our data suggest is aberrantly activated at high temperature in *dyf-1(mn335)* worms. *dyf-1(mn335)* worms remain in the area where they have been placed, independent of their cultivation temperature, suggesting that they lack or cannot act on this

memory. The thermotaxis phenotype of the *dyf-1(mn335)* mutant requires AFD function, suggesting an aberrant sensation of the temperature which is abrogated by AFD mutation.

We demonstrate that loss of integrity of ciliary neurons, particularly AWA and to a lesser extend ASH, causes stabilisation of the UFD substrate. How these neurons regulate UPS-mediated degradation via AFD neurons is not yet clear. The thermotaxis behaviour of the *dyf-1(mn335)* mutant is only partially suppressed by *gcy-8(oy44)* mutation, suggesting that factors other than temperature and aberrant AFD activity contribute to the altered thermotaxis behaviour in *dyf-1(mn335)*. Thermotaxis is a complex behavioural phenotype and was

**Fig. 5 | Neuronal signalling primes the intestine to control UPS activity via calcineurin signalling. a** Part of the calcium signalling pathway Cel04020 (KEGG database) with gene expression changes in *dyf-1(mn335)* compared with WT mapped on it. Pathway nodes correspond to genes coloured in red when gene expression is significantly increased and blue when decreased in *dyf-1(mn335)*, non-affected nodes are coloured in grey, white nodes correspond to genes which were not measured in the microarray experiment. CaN: *tax-6* and *cnb-1*. **b** Protein lysates of indicated worms (and displayed in Supplementary Fig. 6a) were analysed by SDS-PAGE and immunoblotting against GFP and tubulin as loading control. Dots denote presence of indicated mutations, typical result from *n* = 3 biologically independent experiments. **c** Worms expressing the UFD substrate and the mutations indicated by dots were analysed as in (**b**), *n* = 3 biologically independent experiments. **d** same as (**c**) but worms are expressing the indicated mutations or a rescue construct for *daf-16* either in the intestine *(Pges-1::daf-16::GFP)* or in neurons *(Punc-119-daf-*

*16::GFP)*, typical result of *n* = 3 biologically independent experiments. **e** same as (**b**) but indicated mutant worms were grown on control *(ctrl)* or *tax-6 (RNAi)* plates, *n* = 3 biologically independent experiments. **f** Schematic overview of experiments performed. UPS function becomes highly sensitive to the action of calcineurin signalling only when a neuronal signal mediated through *gcy-8* (or absence of *dyf-1*) is present. **g** Indicated mutant worms expressing the UFD substrate were analysed at day one of adulthood by Western blotting against GFP and tubulin as loading control. **h** Quantification of worms displayed in (**g**), *n* = 3 biologically independent experiments. Column graphs: mean ± standard deviation, scatter plots: individual experiments, same colour: same experiment. 1-tailed unpaired Student's *t*- test, *\*p* ≤ 0.05, n.s. *p* > 0.05. *dyf-1(mn335)* versus *dyf-1(mn335); ins-5(tm2560) p* = 0.01114, *dyf-1(mn335) daf-16(mu86)* versus *dyf-1(mn335) daf-16(mu86); ins-5(tm2560)* *p* = 0.03790. Source data are provided as a Source data file.

shown to be influenced by various environmental stimuli such as food, pheromones or humidity[51,62]. For example, sensation of humidity and its associated hygrotaxis requires the function of mechanosensory proteins, FLP, and AFD neurons[51,63]. Also entry into dauer larva, a developmentally arrested stage formed in response to adverse environmental conditions, is influenced by pheromones and sensation of temperature requiring AFD neurons[62]. It will be of interest to determine if the crosstalk between temperature sensation, AFD function and *ins-5* regulation are important to mediate formation of dauer larvae. An inverse relationship between body temperature and lifespan has been found in several organisms[64]. However, it was unclear whether exposure to or perception of higher temperatures underlies the lifespan effect[64]. We found that the *dyf-1(mn335)* mutant exhibits defects in protein turnover at higher temperatures that are reduced by impairment of AFD function, suggesting that higher temperatures and disruption of protein turnover per se are not toxic.

Reduced protein turnover of misfolded proteins by the UPS is generally assumed to result in enhanced protein aggregation as a consequence of elevated misfolded protein levels that escape degradation within the cell[65]. Previous data indicated that AFD activity is important to control chaperone induction and thereby protein misfolding during cellular stress[16,17]. We show here that the *dyf-1(mn335)* mutant possesses less protein aggregates in the intestine, similar to the AFD mutant, whereas its protein turnover defect is suppressed by the same AFD mutation. This indicates that the signalling required to affect UFD stability and protein folding are uncoupled. Consequently, aggregation of polyQ was previously shown to require HSF-1 independent of DAF-16[17]. Worms may have established an ingenious system whereby AFD neurons balance the cellular protein fate to alleviate detrimental health effects if both, protein aggregation and turnover, were increased simultaneously.

Patients suffering from protein aggregation diseases often display an accumulation of protein aggregates in their tissues. Recently, passive or active exercise-related body warming was proposed as a therapy for neurodegenerative diseases[66]. However, patients suffering from diseases such as frontotemporal dementia, Alzheimer's or Parkinson's suffer frequently from aberrant temperature responsiveness[67,68]. Therefore, it will be important to determine how the lack of temperature sensing affects protein turnover and aggregation rates in other organisms, and how passive or exercise-related body warming affects the different proteostasis rates.

## Methods

*Caenorhabditis elegans* strains used in this study are listed in the Supplementary Data 3. Worms were handled according to standard procedures and maintained at 20 °C unless otherwise stated[69].

### Generation of rescue plasmids

The *ins-5* genomic sequence is fused via SL2 splice site to mCherry allowing to measure expression of *ins-5* and *mCherry* from the same

transcript. Oligos used are listed in Supplementary Data 4. The *ins-5* genomic rescue plasmid pTH1219 was created using the NEBuilder® HiFi DNA Assembly kit (NEB, Frankfurt, Germany). The *ins- 5* promoter and *ins-5* gene were amplified with TH2483, TH2484 (2147 bp) and the *ins-5* 3'UTR with TH2489, TH2505 (1182 bp) from *C. elegans* N2 Bristol genomic DNA. The *SL2-NLS-mCherry* fragment was amplified with TH2485, TH2486 (661 bp) and the remaining *mCherry* fragment TH2487, TH2488 (542 bp) from pBalu12[70] and assembled according to the manufacturer's instruction. Subsequently, the 2782 bp *Pins-5-ins-5-SL2-NLS-mcherry* fragment was amplified with TH2483, TH2486 and the 1736bp *mCherry-ins-5-3'UTR* with TH2487, TH2505 and assembled into pBS-*unc-119* (containing genomic *unc-119* as *HindIII/XbaI* fragment) that was linearized with *SalI*. The plasmid contains 1756bp of *ins-5* promoter region and 1182 bp of *ins-5* 3'UTR. The *ins-5* intestinal rescue plasmid pTH1707 was created using the NEBuilder® HiFi DNA Assembly kit. 2030 bp promoter sequence of *ges-1* (amplified with TH3147 and TH3148 from genomic N2 Bristol DNA) as well as 410 bp sequence of *ins-5-SL2* amplified with TH3149 and TH3146 from pTH1219 were inserted into pTH1219 that was cut with with *XhoI* and *HpaI*. The *ins-5* neuronal rescue plasmid pTH1720 was created using the NEBuilder® HiFi DNA Assembly kit. The 2165 bp promoter sequence of *unc-119* (amplified with TH3159 and TH3144) and 410 bp sequence of *ins-5-SL2* amplified with TH3145 and TH3146 were inserted into pTH1219 that was cut with with *XhoI* and *HpaI*. *Punc-119* and *Pges-1* promote pan-neuronal and intestinal expression, respectively.

### Generation of transgenic *C. elegans*

Generation of transgenic worms was performed as described[21]. In brief, microparticle bombardment into strains carrying the *unc-119(ed4)* mutation was done using the BioRad Biolistic PDS-1000/HE (BioRad, Feldkirchen, Germany) equipped with a heptashot adapter according to the manufacturer's instruction with 27 inches of Hg vacuum and a 1350 p.s.i. rupture disc. Per bombardment, about 1 mg of 0.3–3 µm gold beads (Chempur, Karlsruhe, Germany) were coated with 7 µg linearised plasmid DNA or 10 µg fosmid DNA. Animals were allowed to recover for 1 h at room temperature and then transferred to 90 mm plates seeded with OP50. Animals were screened for integration after three weeks at 25 °C for rescue of the *unc-119(ed4)* phenotype.

### RNAi

RNA interference was performed using the feeding method[71]. HT115 bacteria carrying the vector with the gene of interest were either taken from the RNAi Collection (Ahringer) or the ORF-RNAi Resource (Vidal) (Source BioScience). As control, HT115 bacteria transformed with the empty pPD129.36 vector was used for feeding. Typically, worms were grown at 22 °C and treated with RNAi from L1 or L3 stage and the phenotype observed during day one of adulthood unless otherwise stated. For enhanced RNAi efficiency in neuronal tissues the *nre-*

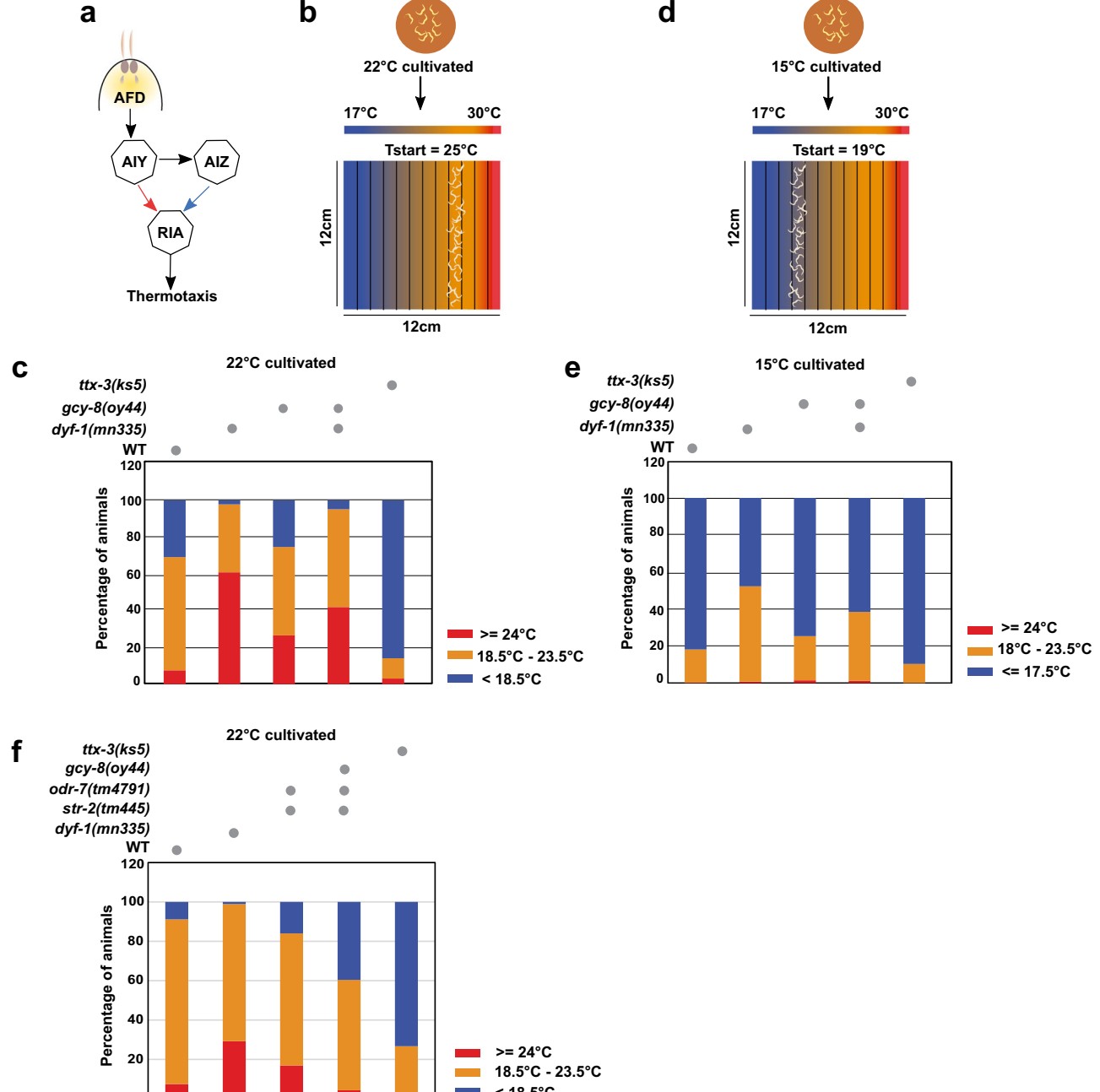

**Fig. 6 | *dyf-1(mn335)* mutant exhibits impaired thermotaxis behaviour. a** A simplified model of thermotaxis neural circuit. Temperature is sensed by the thermosensory neuron AFD. The AIY-RIA and AIZ-RIA circuit promotes the movement of animals to a higher temperature (thermophilic behaviour) and lower (cryophilic behaviour) temperature than the cultivation temperature, respectively. The RIA interneuron integrates the thermal signals and drives thermotaxis behaviour. **b** The linear thermal gradient setup established on a 12*12 cm agar surface. Tstart indicates the temperature at which the animals were placed on the linear gradient after they were cultivated at 22 °C. **c** Negative thermotaxis behaviour of animals cultivated at 22 °C after 60 min are shown. The percentage distribution of animals in the specified temperature ranges are indicated. Number of animals are WT 552, *dyf-1(mn335)* 1117, *gcy-8(oy44)* 759, *dyf-1(mn335); gcy-8(oy44)* 810 and *ttx-*

*3(ks5)* 815. **d** Linear thermal gradient setup similar to (**b**), but worms were cultivated at 15 °C, and positioned at Tstart of 19 °C. **e** Negative thermotaxis behaviour of animals cultivated at 15 °C. T start = 19 °C. Number of animals are WT 569, *dyf-1(mn335)* 691, *gcy-8(oy44)* 497, *dyf-1(mn335); gcy-8(oy44)* 510 and *ttx-3(ks5)* 581. For detailed statistics see Supplementary Tables 2 and 3. 3 biologically independent experiments (**c, e**). No gradient is displayed in Supplementary Fig. 7a–d. **f** Negative thermotaxis behaviour of animals cultivated at 22 °C after 60 min are shown. Number of animals are WT 654, *dyf-1(mn335)* 831, *str-2(tm445); odr-7(tm4791)* 1209, *gcy-8(oy44); str-2(tm445); odr-7(tm4791)* 720, and *ttx-3(ks5)* 978. 3 biologically independent experiments. For detailed statistics see Supplementary Table 4. Source Data are provided as a Source Data file.

*1(hd20); lin-15b(hd126)* strain background was used[72]. When RNAi of *tax-6* was performed to address *gcy-8(oy44)* neuronal effect or after *hecd-1(RNAi)*, no RNAi-sensitive background was used (Fig. 5e, Supplementary Fig. 6b).

## Lifespan analysis

Lifespan was studied with PP3336, PP3337, PP3338, and PP3339, all of which resulted from outcrossing PP2079 with N2 Bristol. Experiments were performed in four replicates at 22 °C. For each lifespan, 50 age-

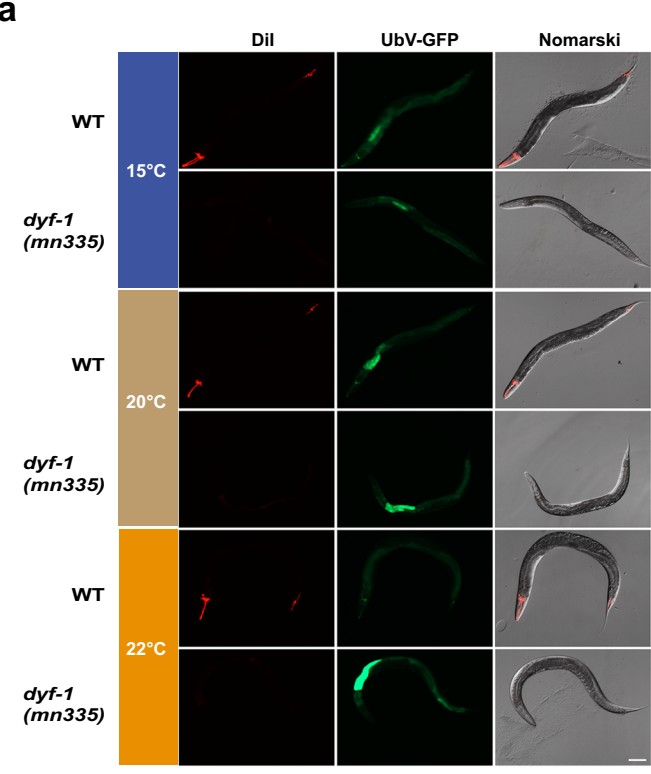

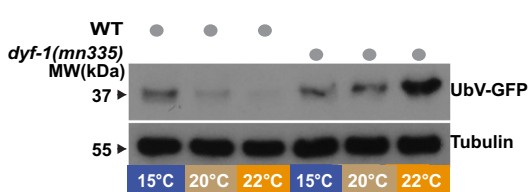

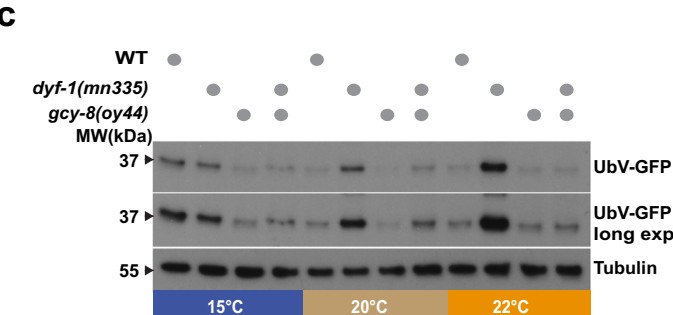

**Fig. 7 | Sensory temperature perception adjusts protein turnover.**
**a** Fluorescence and Nomarski images of worms expressing the UFD substrate (UbV-GFP) in WT or *dyf-1(mn335)* mutant worms grown at different temperatures as indicated. DiI denotes ciliated amphid and phasmid neurons. $n = 3$ biologically independent experiments. Scale bar 100 μm. **b** Protein lysates of worms displayed in (**a**) were analysed by SDS-PAGE and immunoblotting against GFP and tubulin as loading control. $n = 3$ biologically independent experiments. **c** Lysates of WT or *dyf-1(mn335), gcy-8(oy44)* or *dyf-1(mn335); gcy-8(oy44)* mutant worms grown at indicated temperatures were analysed as in (**b**) $n = 3$ biologically independent experiments. Long exp. = long exposure of the blot. Source Data are provided as a Source Data file.

synchronised, day 1 adult hermaphrodites per strain were transferred to fresh NGM agar plates, corresponding to day 0 in the lifespan experiment. Animals were transferred to new plates daily to separate adults from offspring and prevent starvation. Survival was assessed daily by monitoring pharyngeal pumping and touch-provoked movements until death. Worms that crawled up the plate walls and desiccated or exhibited a 'protruding vulva' or showed 'bagging' were excluded from the study. Statistical details for all lifespan experiments are shown in Supplementary Table 5.

**Sample collection and Western blotting**
Western blotting was performed essentially as described[21]. To determine the phenotype in mutants lacking distal cilia formation, worms were grown at 20 °C until adulthood, eggs synchronized by bleaching to avoid contamination and eggs immediately placed on NGM containing OP50 to avoid starvation or overcrowding at 22 °C unless otherwise stated. After 40 hours 100 young L3 worms were selected and manually transferred (2 × 50 worms) to fresh NGM-OP50 plates for another day at 22 °C. Hermaphrodite worms were collected for analysis at early day one of adulthood. For preparation of whole worm lysates, 100 worms were collected in 1x M9 buffer, settled on ice, the supernatant removed and the worm pellet suspended in 50 μl 2x SDS loading buffer. To investigate the polyubiquitylation status of UbV-GFP the worm pellet was lysed in SDS sample buffer containing 10 mM n-ethylmaleimide (NEM) to block deubiquitylation enzymes. Subsequently, samples were boiled at 95 °C for 4 min, sonicated 2 × 10 sec at 60% amplitude and again boiled at 95 °C for 4 min. Samples were then spun at 18,000 g for 2 min at room temperature and the supernatant applied to NuPAGE® 4–12% Bis-Tris SDS-gels using the NuPAGE® MES SDS running buffer (ThermoFischer Scientific, Germany). Samples were transferred using a semi-dry blotting system (Bio-Rad, Trans-Blot Turbo) with NuPAGE® transfer buffer. Antibodies were diluted (anti-

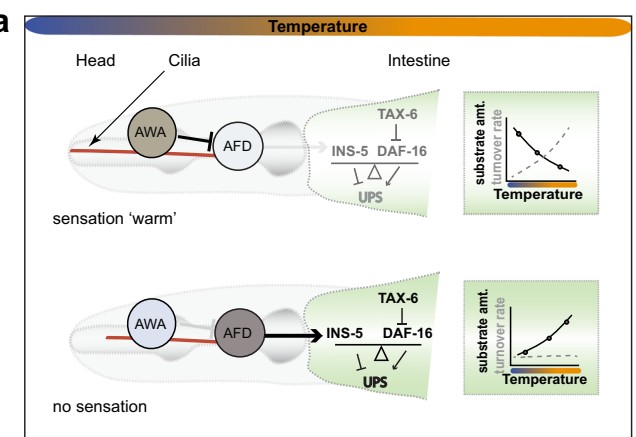

**Fig. 8 | Sensory temperature perception and proteostasis regulation. a** In WT worms, external stimuli, such as warm temperature, are perceived and processed normally via sensory neurons such as AWA (and possibly other sensory neurons, including ASH), reducing AFD signalling (sensation 'warm'). The red lines schematically show the length of all cilia, which are longer in WT, whereas the distal segments are absent in the *dyf-1(mn335)* mutant. UPS function in the intestine does not primarily depend on INS-5 or calcineurin signalling. Thus, WT worms increase their protein degradation in response to warm temperature (dotted line in the graph), resulting in decreased amounts of proteasomal substrates (substrate amt. dark line) at higher temperature (substrate amt. dark line). *dyf-1(·)* mutant worms exhibit defects in distal cilia formation and are unable to properly sense ambient temperature (no sensation) due to decreased AWA function creating a neuronal signal that involves the aberrant function of the AFD neurons which fail to sense the ambient temperature correctly (indicated by black arrow). This signal includes DCV secretion and the neuropeptide FLP-3 that blocks the UPS in the intestine. The induced AFD signal increases INS-5 to reduce UPS function. In parallel, *tax-6* loss-of-function is tipping the balance towards the action of DAF-16 to control the UPS. Hence, *dyf-1(·)* worms fail to adjust their protein degradation rate (dotted grey line in the graph) due to loss of temperature sensation and accumulate proteasomal substrates (dark line in the graph).

GFP 1:5000 (Clontech), anti-tubulin 1:5000 (Sigma), anti-mCherry 1:2000 (Abcam), anti-mHRP 1:10,000 (Jackson Immunoresearch, anti-mouse IRDye 1:10,000 (LI-COR) in 5% dried milk/ PBS/ 0.1% Tween 20 (for regular ECL detection) or Roti®-Block (Carl Roth (for visualization with Odyssey). Visualization of fluorescent signals was performed in early experiments using the Odyssey scanner (LI-COR, Bad Homburg, Germany) and the Image Studio Lite v4.0 software. However, in most cases the chemiluminescent detection with regular self-made ECL detection solution was used as this was providing a robust correlation between the GFP signal observed in worms by fluorescence microscopy with the GFP signal detected by Western blotting. Blots were imaged using X-ray films. All information regarding resources can be found in Supplementary Table 6, all uncropped blots are provided in the Source Data file accompanied with this manuscript.

### DiI staining
For staining, hermaphrodite worms were washed from plates with 1x M9 buffer, transferred to an Eppendorf tube and settled on ice, the supernatant discarded and 1 mL of the DiI (2 μg/mL) staining solution was added and worms stained for 30 min with continued end-over-end rotation at room temperature. Worms were settled on ice, washed three times with 1x M9 and allowed to recover for 1 h on NGM plates seeded with OP50 before imaging.

### In vivo imaging
In vivo imaging was performed as described recently[21]. Worms were imaged on a 4% agarose pad containing 30 mM sodium azide using an Axio imager M1(Carl Zeiss Microscopy GmbH, Jena, Germany)

equipped with Axiocam 503mono and Zeiss2.3 pro software (Carl Zeiss Microscopy GmbH). When worms were imaged on plates, the plates were precooled on ice for 5 min to immobilise the worms and imaged using Leica M165FC equipped with Leica DFC340FX camera and Leica application Suite 3.3.1 software or Axiozoom V16, (Carl Zeiss Microscopy GmbH) equipped with Axiocam 506mono and Zeiss 2.3 software (Carl Zeiss Microscopy GmbH).

For confocal imaging of ciliary structures, day 1 adult hermaphrodites were cultured at 22 °C and immobilised with 25 mM levamisole on a 3% agarose pad. Worms were imaged using LSM980 Airyscan 2 equipped with Plan-Apochromat 63x /1.4 oil DIC and ZEN Connect Modul (Carl Zeiss Microscopy GmbH). The presence of the *dyf-1(mn335)* mutation was confirmed by DiI staining. Image J was used for image processing and analysis.

### Mass spectrometric analysis
Per condition 60 young adult hermaphrodite worms (PP563, PP915, PP1058, PP1716) were grown at 22 °C and collected in 20 μL 1x M9 buffer, the buffer removed and worms shock frozen in liquid nitrogen. A total of 3 independent experiments were performed with all strains. Worms were lysed in 40 μL 6 M Urea/2 M Thiourea buffer with sonification at 4 °C with 30 s intervals for 10 min. Samples were reduced in 50 mM DTT for 30 min at RT followed by alkylation using 13 mM iodoacetamide for 20 min in the dark, before digestion with 0.4 μg lysyl-endopeptidase for 3 h at RT. The lysates were diluted with 50 mM ammonium bicarbonate to get a final urea concentration of 2 M and digested with 2 μg trypsin. One day later, the digest was stopped by acidification with an equal amount of buffer C (5% acetonitrile, 1% TFA) before peptides were cleaned up using stage tip extraction (Empore Disks, Sigma Aldrich, Taufkirchen, Germany). For separation of the peptides the stage tips were prepared by washing with methanol (2600 rpm, 2 min) followed by washing with elution buffer (80% acetonitrile, 0.1% formic acid) and two times with washing buffer (0.1% formic acid in ddH$_2$O). The samples were loaded on the stage tip, cleared once more with washing buffer and dried until further processing.

### LC−MS/MS
Eluted peptides were separated using a binary buffer system of A (0.1% (v/v) formic acid in H$_2$O) and B (0.1% (v/v) formic acid in 80% acetonitrile) on a nanoEasy HPLC system (Thermo Fisher Scientific). A linear gradient from 7–35% B was applied within 220 min followed by 95% B for 10 min. Prior to the next run, the column was re-equilibrated to 5% B for 10 min. The 50 cm column (75 μm ID) was packed in-house with 1.9 μm diameter C18 Reprosil resin (Dr. Maisch GmbH, Ammerbuch-Entringen, Germany). A custom-made column oven was used to control the temperature to 40 °C. The HPLC was coupled via a nano-electrospray ionization source (Thermo Fisher Scientific, Bremen, Germany) to the quadrupole-based QExactive Plus benchtop mass spectrometer (Thermo Scientific, Bremen, Germany). MS spectra were acquired using 3e6 as AGC target at a resolution of 70, 000 (200 m/z) in a mass range of 350–1650 m/z. A maximum injection time of 60 ms was used for ion accumulation. MS/MS events were measured in a data-dependent mode for the 10 most abundant peaks (Top10 method) in the high mass accuracy Orbitrap after HCD (Higher energy C-Trap Dissociation) fragmentation at 25 collision energy in a 100–1650 m/z mass range. The resolution was set to 17, 500 at 200 m/z and the injection time was set to 60 ms. Acquired raw files were correlated to the Uniprot *Caenorhabditis elegans* reference proteome (Proteome ID: UP000001940, canonical protein sequences: 26196, downloaded 03.2015) using MaxQuant (version 1.5.3.8)[73] and the implemented Andromeda search engine[74]. Searches were performed with tryptic digestion specificity allowing two missed cleavages and a mass tolerance of 4.5 ppm for MS and 6 ppm for MS/MS spectra. Carbamidomethyl at cysteine residues was set as a fixed modification and

oxidation at methionine, and acetylation at the N-terminus were defined as variable modifications. The minimal peptide length was set to seven amino acids, and the false discovery rate for proteins and peptides to 1%. The mass spectrometry proteomics data have been deposited to the ProteomeXchange Consortium via the PRIDE[75] partner repository with the dataset identifier PXD016676.

## RNA extraction

RNA extraction was performed as described recently[15]. Age-synchronised day 1 adult hermaphrodite worms grown at 22 °C (200–300 if RNA was prepared for microarray, 200 for qRT-PCR) were washed two times with M9 buffer, resuspended in 1 ml TRIzol reagent (Ambion) and frozen at −80 °C for at least 1 h. Worms were homogenised by addition of 200 μL Zirconia beads, followed by tissue disruption with the Precellys 24-Dual cell homogeniser (VWR, Germany). Next, 1-bromo-3-chloropropane was added to the samples followed by vigorous mixing and phase separation via centrifugation. The aqueous phase was used to isolate total RNA with the RNeasy Mini Kit (Qiagen, Hilden, Germany) according to manufacturer's instructions. cDNA synthesis was performed with 200 ng total RNA (for microarray) or 1 μg (for qPCR) using the High-Capacity cDNA Reverse Transcription Kit (Applied Biosystems, Germany). Gene expression levels were measured via quantitative real time PCR (qRT-PCR) with the Luna Universal qPCR Master Mix (New England Biolabs) and the Bio-Rad CFX96 Real-Time PCR Detection System. Data were analysed using the Bio-RadCFX Manager 3.0 software. Actin served as internal control and primer efficiencies were determined by creation of a standard curve. For qRT-PCR typically three independent experiments were performed in triplicates. For primers see Supplementary Data 4, for key reagents, and resources see Supplementary Table 6).

## Microarray analysis

Microarray analysis was described recently[15]. Expression profiling of the PP896 dyf-1(mn335) mutant versus wild-type N2 Bristol was done using the Affymetrix C. elegans Gene Array 1.0. To this end, total RNA of around 200 − 300 age-synchronised day 1 adult hermaphrodites was extracted as described above. Microarray analysis was performed using four replicates per strain and was processed by the Cologne Center for Genomics (CCG, Cologne, Germany).

## Cycloheximide treatment

Hermaphrodite worms were grown on NGM plates seeded with OP50, bleached and embryos transferred to fresh NGM plates seeded with OP50 and grown at 22 °C till L3 stage. Cycloheximide plates were prepared by spreading 50 μL of 50 mg/mL cycloheximide (Sigma) in ethanol stock solution on a small petri dish containing 5 mL NGM using a spatula, as control only ethanol was added to the plates and all plates incubated at room temperature for 30 min. Plates were then seeded with OP50 bacteria overnight at 37 °C. At least 600 early L3 worms were transferred to new NGM-OP50 plates until early day 1 of adulthood. 100 day one adult worms were then transferred for 0 h, 1.5 h or 3 h to NGM plates containing freshly prepared cycloheximide or control plates. Worms were then collected and analysed by Western blotting.

## PolyQ aggregation analysis

For quantification of polyQ aggregates, hermaphrodite worms were grown as above and the number of aggregates per worm counted by live imaging. Worms counted were transferred to new plates to avoid miscounting. Worms containing more than 80 aggregates were counted as 80.

## Thermotaxis assay

A 12*12 cm plate was used for the linear thermotaxis assay. The thermotaxis plate was divided into twelve regions using a marker pen.

Assay plates were filled with 15 mL of cool (≤55 °C), liquid media (Agar (17 g), NaCl (3 g) in 975 mL ddH$_2$O, CaCl$_2$ (1 mL of 1 M solution), MgSO$_4$ (1 mL of 1 M solution), and KH$_2$PO$_4$ (25 mL of 1 M solution, pH 6.0)). The plates were dried for 2 h at RT prior to use. The linear gradient was established on the 12*12 cm plate containing the liquid media by keeping the plate on an aluminium slab connected by two water baths kept at 8 °C and 55 °C. After 30 min a gradient ranging from 17 °C to 30 °C was reached. Once the assay plates reached this thermal gradient, synchronised, well-fed young adult worms from the growth plates were added to the thermotaxis plate. To this end, day 1 adult hermaphrodite worms were transferred in ddH$_2$O to a 1.5 mL centrifuge tube. The tube was spun at 550 rpm for 30 seconds to pellet the animals. The animals were washed twice in ddH$_2$O and later suspended in 100 μL. The animals were then transferred to a narrow starting zone (T start) of each assay plate. The droplet was dispersed across the T start zone using a tissue paper. The assay clock started as soon as animals begun to crawl away from the starting zone and the assay conducted for 60 min. The temperature of each region was measured using a hand-held laser thermometer after 60 min of the experiment. The thermotaxis plate was then moved to an ice bucket and transferred to a 4 °C for 60 min to paralyse the animals. The terminal points of the animals were manually counted in each of the twelve regions to calculate the distribution of the animals in a specific temperature range. Worms which crawled up the plate walls and dried up during the 60 min of the assay were censored from the study.

## Statistical analysis and data visualisation

Statistical analysis of Western blots was performed according to[76]. For quantification of UFD stability, the GFP signal was normalised by the tubulin levels and the ratios compared with the wild-type which was set to one, or to dyf-1(mn335) when suppression was analysed. Typically, an unpaired two-tailed Student's t-test was performed using Microsoft Excel 2016 software (Microsoft). In cases where the direction of the distribution was already known from previous experiments, a one-tailed Student's t-test was performed. When several mutants were compared with each other a one-way ANOVA test was performed followed by post-hoc comparison's tests as stated in the Figure legends using Graph Pad Prism 7 software. Results are typically provided as mean ± standard deviation (if not otherwise indicated) where n refers to independent experiments. For the thermotaxis experiments, chi-square values were calculated using Microsoft Excel 2016 and Graph Pad Prism 7 software. For statistical analysis of lifespan, Graph Pad Prism 7 software was used. For survival tests, significance was determined using the Log-rank test (Mantel-Cox) and the unpaired Student's t-test. Statistical analysis of microarray and proteomics data and visualisation of results was conducted in R programming environment[77]. For microarray data analysis R package oligo v1.42.0[78] was used to read Affymetrix.CEL files in R environment and to perform Robust Multi-Array Average (RMA) normalisation[79] using background adjustment, quantile normalisation, and median polish for Single-Channel. Next, single channel differential expression analysis comparing dyf-1(mn335) to wild-type N2 was performed with limma v3.34.9 R package[80]. Gene Set Enrichment Analysis (GSEA) algorithm was used as described in[81,82]. GSEA determines whether an a priori defined set of genes (e.g. biological pathway) shows statistically significant association with the phenotype of interest. GSEA implemented in R package ReactomePA1.34.0 was used to analyse the microarray data with Reactome pathways. KEGG calcium signalling pathway Cel04020 was overlaid with expression changes in dyf-1(mn335) compared with wild-type and visualised using pathview v1.18.2 R package[83].

For Mass Spectometry data analysis DEP package v1.8.0[84] was used for label-free quantification (LFQ)-based analysis of MaxQuant output (peptides.txt file). The function uses protein-wise linear models combined with empirical Bayes statistics to perform moderated t-test with the limma package for differential expression analysis[80]. p values

of DE tests were FDR adjusted for multiple testing and resulting $q$ values were used for prioritising proteins of interest. The data was background-corrected, normalised by variance-stabilising transformation (*vsn*), missing values were imputed and differential enrichment analysis was performed.

### Reporting summary

Further information on research design is available in the Nature Research Reporting Summary linked to this article.

## Data availability

Source data are provided with this paper. All data needed to evaluate the conclusion are available in the main text or the supplementary data files provided with this manuscript. Mutant strains are available upon request from the authors and/or CGC. The microarray data generated in this study have been deposited under the accession code GSE142371 in the Gene expression omnibus database (GEO, NCBI) and are publicly available: https://www.ncbi.nlm.nih.gov/geo/query/acc.cgi?acc=GSE142371. The Gene ontology database is available via: http://geneontology.org/, the Reactome database is available via: https://bioconductor.org/packages/release/bioc/html/ReactomePA.html. The Uniprot reference *C. elegans* proteome database is available at: https://www.uniprot.org/proteomes/UP000001940. The proteomics data have been deposited with the identifier PXD016676 via the ProteomeXchange database and are publicly available: https://www.ebi.ac.uk/pride/archive/projects/PXD016676. Source data are provided with this paper.

## Code availability

Code to reproduce the data is freely available on the authors github repository[85]: https://github.com/nevelsk90/Celegans_sensingDyf1.git or via the DOI identifier: https://doi.org/10.5281/zenodo.7007326.

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

## Acknowledgements

We thank O. Hobert for valuable advice on the neuronal identity assay, R. Morimoto for strains. S. Greco-Torres and G. Vopper for technical assistance, C. Ahyoud for generating strains, the Cologne Centre for Genomics (CCG) for microarray analysis, H. Nolte (CECAD proteomics facility) for proteomics analysis, SunyBiotech for generating the GFP::C05D11.5 strain, L. Leson for help with setting up the thermotaxis gradient device. *C. elegans* strains were kindly provided by the *Caenorhabditis* Genetics Center (funded by the NIH Office of Research Infrastructure Programs (P40 OD010440), and the Mitani lab. We also thank A. Andersen (Life Science Editors), C.E. Kutzner, S. Ravanelli, and D. Vilchez for critical reading of the manuscript. This work was funded by the Deutsche Forschungsgemeinschaft (DFG, German Research Foundation) under Germany´s Excellence Strategy – EXC 2030 – 390661388 and by the European Research Council (ERC-CoG-616499) to T.H. Diese Arbeit wurde von der Deutschen Forschungsgemeinschaft (DFG) im Rahmen der deutschen Exzellenzstrategie – EXC 2030 – 390661388 und vom Europäischen Forschungsrat (ERC-CoG-616499) an T.H. gefördert. K.L.V. received support by the Cologne Graduate School of Aging Research.

## Author contributions

A.S. and T.H. designed the study and supervised the data interpretation, A.S., K.L.V., V.K., Q.L., J.L., L.K., F.F. performed individual experiments and analysed the data, A.S. and T.P. analysed the microarray data and mass spectrometry, A.S., K.L.V., and T.H. wrote the paper.

## Funding

## Competing interests

The authors declare no competing interests.
