## [Peer Review File · Nature Communications]

Thermosensation in *Caenorhabditis elegans* is linked to ubiquitin-dependent protein turnover via insulin and calcineurin signallingREVIEWER COMMENTS

Reviewer #1 (Remarks to the Author):

This is a beautiful and very important paper. It was a pleasure to read (and to review). Very briefly, what Segref et al. show is that ciliated neurons (and they also dig into the role of a few specific neurons, AWA and AFD) control ubiquitin dependent protein turnover non-cell autonomously. They find also numerous mechanistic details regarding this systemic effect, notably the involvement of FLP-3, INS-5, Tax-6 and Daf-16 (and the site of action of these molecules). These results would be of interest to many scientists, not only *C. elegans* and proteostasis experts, as they shed light on "mind and body" communication, which is a very hot topic. It has been shown in the past that AFD controls the heat shock the worms and has different non cell autonomous effects, but this paper add many novel insights.

The results are very convincing and every conclusion is validated in a number of different ways/techniques. The paper combines the powerful genetics of *C. elegans* (many mutants are examined, many genes are endogenously tagged) and also excellent biochemistry work (including, for example, validation using mass spectrometry of endogenous targets and many other methods). I will be very happy to see this published in Nature Comm and have just a few very small comments:

"To test if these peptides mediate the neuronal signalling and resulting protein degradation defects in *dyf-1(mn335)* worms, we combined RNAi and genetic analyses. Indeed, depletion and mutation of the neuropeptide FLP-3 and the insulin like peptide INS-5 restored UPS-dependent degradation in *dyf-1(mn335)* worms (Fig. 4 C and D, Fig. S5 C and D, and Table S3)."

– Until I looked it up in the methods I wondered whether this means that FLP-3 in fact functions in non-neuronal tissues, because RNAi doesn't work well in neurons. When I checked the methods I found out that they (correctly) did RNAi in mutants that sensitise neurons for RNAi. I think they should say it clearly in the main text as well.

Small typo, the references in this sentence are unformatted: "Thermosensory AFD neurons regulate cellular HSR upon acute heat shock, which triggers organism-wide adaptation of proteostasis (Prahlad and Morimoto, 2008, 2011)"

In my opinion the paragraph "Ciliary mutant worms display aberrant thermosensation" is not perfectly placed in terms of the paper's "flow". It's not clear why it's discussed when it is discussed. Perhaps if they somehow discuss their hypothesis regarding the mechanism earlier it would be clearer. Could you somehow say at the beginning of the paragraph that you hypothesize that AFD signalling inhibits UFD substrate turnover at relatively low temperatures, which is counteracted by sensory neurons at higher temperatures?

Reviewer #2 (Remarks to the Author):

My comments are also included in the document attached.

What are the noteworthy results?

This article helps define a cellular and genetic pathway that control protein turnover in the intestine of *C.elegans*. Importantly, the pathway is initiated by the loss of function in ciliated sensory neurons, which leads to transmission, likely via a peptide signal, to the intestine, where daf-16-regulated protein turnover is inhibited.

Particularly important results include:

- Identification of ciliated neurons as regulators of protein turnover (Figure 1) in the intestine, with potential other sites of regulation (hypodermis, Fig S1).
- Finding that the signaling from AFD neurons that lack functional cilia (*dyf-1* mutants) is dependent upon the AFD-specific guananyl cyclase, *gcy-8*, and is not a result of increased protein misfolding in the intestines of *dyf-1* animals, generally (Figure 2).
- Proteomic and biochemical analysis showing that endogenous proteins (not just the engineered UbV-GFP substrate) are targets of the ubiquitin/proteasome-degradation pathway that is disrupted by the loss of *gcy-8* signaling (Figure 2, S2), and that the same substrates are not upregulated transcriptionally (Figure 3).
- Identification of *ins-5* is upregulated in response to loss of AFD function, and intestinal expression of *ins-5* negatively stabilized UbV-GFP in *dyf-1* mutants (Figure 4).
- Determining that intestinal, but not neuronal, calcineurin (*tax-6*) and daf-16 signaling mediate UbV-GFP degradation (Figure 5).
- Thermotaxis mediated by ciliary neurons requires *dyf-1* and thermotactic defects can be partially rescued by blocking signaling from AFD via *gcy-8*.
- Temperature sensing by ciliated neurons helps control temperature-dependent proteostasis in other tissues.

Will the work be of significance to the field and related fields? How does it compare to the established literature? If the work is not original, please provide relevant references.

·Yes, this is a significant study. It expands on a growing body of research that has described the connections between neurons and intestinal cells in *C. elegans*, specifically regarding protein homeostasis and stress (Taylor and Dillin, 2013, Imanikia, et al., 2019). Of particular interest, this

paper describes an how an environmental stimulus (temperature fluctuation) can be sensed and transmitted to other tissues. There is a candidate for a signaling molecule between the two cell types (*flp-3*), and while this paper does not specifically probe stress responses, the change in protein degradation may well indicate a role for proteostatic stress or the UPR downstream of attenuated neuronal function.

Does the work support the conclusions and claims, or is additional evidence needed?

·In general, there is clear evidence for the claims made. However, I did not find direct evidence that the AWA pathway is in the same genetic pathway as AFD, as opposed to working in parallel. One way to probe this connection could be through an *odr-7*; *gcy-8* double mutant, which should phenocopy the *dyf-1*; *gcy-8* double mutant (Figure 2D). Additional evidence could be provided by investigating whether the downstream effectors *ins-5* and *flp-3* are upregulated in *odr-7* mutants, as they are in *dyf-1* mutants.

·These two pieces of evidence would more precisely implicate AWA as sending signals through AFD, in the absence of AWA function. In addition, a cell-specific rescue of *dyf-1* in AFD (*Pgcy-8::Dyf-1*) could be used to show that the *dyf-1* phenotype is not due to AFD structural abnormalities (that is, lack of cilia).

The authors acknowledge that ciliated neurons other than AFD contribute to the thermotaxis and substrate stabilization phenotypes shown in Figures 6 and 7. Using a cell-specific rescue of the cilia in AFD would strengthen this hypothesis and provide support for the model in Figure 8.

·It would also be helpful to include images of any expected or unexpected ciliary defects in AFD and AWA in the *dyf-1*, *odr-7*, and *gcy-8* mutants. *Pgcy-8::GFP* and *Podr-10::GFP* animals are available (Ou, et al., 2007 and others) and could be used for confocal imaging. This is true for all of the ciliated neurons tested in Figure 2, but AFD and AWA are most central to this article.

·The authors show that *dyf-1* mutants do not exhibit higher levels of misfolding of a model protein (*Htt, Q44::GFP*). However, given the greatly increased level of substrate accumulation and ubiquitylation, it would be interesting to know if the *dyf-1* and/or *odr-7* (AWA-specific) mutants have any increase in UPR or other proteostatic stress responses. *hsp-4* or *hsp-16.2* reporter strains could be used to provide evidence of UPR or cytosolic protein-based stress responses. (Alternatively, detection of *xpb-1s* mRNA could be used to determine if the UPR is activated in these mutants.)

Are there any flaws in the data analysis, interpretation and conclusions? Do these prohibit publication or require revision?

·Because AFD is a crucial thermosensory neuron, it the thermotaxis data would be strengthened by an AFD-specific ciliary defect. As it is, the thermosensory data describes a defect that likely has contributions from a number of cells, being able to parse those from the AFD defect itself is important. An example experiment could use the cell-specific rescue described above to drive DYF-1 specifically in the AFD. I believe that that this point should be addressed in the text and/or with data prior to publication.

Is the methodology sound? Does the work meet the expected standards in your field?

·Yes, generally. The methodology is sound and uses the correct statistical and imaging/imaging-quantification protocols.

·Exception: Fig. 1F shows a blot with ladderized UbV-GFP, which is described as Ubn-GFP. However, the blot uses (according to the figure legend) an anti-GFP-antibody. Please repeat this blot using anti-Ub to confirm that these are Ub conjugates (which they very likely are). This is important given that it is unclear why this blot, but no others in the article, show the ladderizing, even in long exposures. I presume that it is due to cropping of the other blots, but it should be clarified in the figure legend and/or the text.

Is there enough detail provided in the methods for the work to be reproduced?

·Yes, when combined with Table S8.

·Exception: the gene *ivd-1* (Fig. 1F) is not described anywhere in the manuscript, including the figure legend or the strains used. This should be corrected.

Other comments:

Authors should acknowledge the work of R. Taylor regarding neural->intestine signaling, especially in the light of protein homeostasis.

Throughout the manuscript ciliary defect/variability and neural function can be hard to parse. This is important especially when discussing AFD function in *dyf-1* vs *gcy-8* mutants, which both affect AFD, but in different ways. (See line 125 for an example)

Fig. 1F: The gene *ivd-1* (and its mutant) is not included in any materials or text.

Line 130-135: Please state explicitly that this experiment is a control to show that *dyf-1* loss of *dyf-1* or *gcy-8* does not result in increased protein misfolding, leading to substrate accumulation. It is currently not clear from the text how these results are important, though it is a useful control.

Fig. 3. There is no explanation of the worm schematics in the figure legend. Please add a description to clarify.

Fig. S3A: for clarity, I suggest that the WT and *dyf-1* conditions be transposed (they are currently displayed *dyf-1* then WT).

Fig. S3F-H: Please use the same formatting for Figures F and H.

Line 155-156: I believe that this sentence should read "ANMT-2-GFP and C05D11.5 are NOT degraded but eh 26 S proteasome upon sensory defects." If my reading of the is not correct, please clarify this result more fully.

Line 163: It is not entirely clear why these two genes were chosen for follow-up. Please include a clarification of why these, but not other candidates, were pursued.

Line s 175-189: Given that *flp-3* may be a crucial link between the neural and intestinal systems, it is not clear why a *flp-3* mutant was not further investigated, though *ins-5* was.

Figure 5A. Please indicate more clearly in the figure that CaN is orthologous to TAX-6. It noted in the legend and the text, but not on the figure itself.

Line 221: Please clarify if or that AFD itself is also affected by the ciliary defect.

Figure 8: This model is somewhat confusing as it shows AWA lacking cilia in both panels. Please clarify if both cells are meant to have cilia, or if the red line represents all neural cilia, in general. Moreover, although the effect of the *odr-7* mutation does stabilize the UbV-GFP substrate, its effect is not as complete as the *dyf-1* mutants', so it may not be a single cell that is blocking the downstream signaling from AFD. The figure legend does not mention AWA; the authors should more fully explain how AWA functions in their model.

Lines 331-336: This paragraph is something of a non sequitur but could be improved by either including the term “in other organisms” in the final sentence, or by suggesting if/how *C. elegans* models might be of use in identifying causal links between disease phenotypes and the cellular protein turnover.

Reviewer #3 (Remarks to the Author):

This study by Segref et al demonstrates that the lack of sensory perception, via AFD neuronal signalling activity, regulates ubiquitin-dependent protein turnover in the intestine. The authors identify INS-5 (insulin-like signalling) and TAX-6 (calcineurin signalling) as mediators of this neuron-to-gut communication, with negative regulation of the FOXO transcription factor DAF-16 regulating UPS activity. Because this signalling requires the thermosensory AFD neurons, the study also provides evidence that UFD substrate is turned over in a temperature dependent manner.

Overall this is a very elegant and well controlled study. It provides an important advance in the field of organismal proteostasis, that now couples neuronal perception with protein turnover in the intestine.

My only major question here is how this influences survival of *C. elegans* during thermal stress conditions. This is not addressed in the study at all, and this is known from the original study by Prahlad et al, *Science* 2008, that used the *gcy-8* mutant. Another recent study reported that neuronal HSF-1 induces fat remodelling in the gut, that is required for *C. elegans* survival at warm temperatures (Chauve et al., 2021, *Plos Biol* 19(11): e3001431). The question now is, how does neuronal perception – or the lack thereof with the *dyf-1* mutant affect survival to thermal challenges? In other words, is the *dyf-1* mutant thermosensitive?

Specific points:

Figure 2C: You mention in the text that protein degradation requires AWA dependent signalling in particular, but UbV-GFP levels are also significantly increased in ASH. Is ASH dependent signalling not important for protein degradation? This point should be at least addressed in the Discussion and commented on.

Figure 2F: The *dyf-1* mutant shows less Q44 aggregation in the intestine, similar to a *gcy-8* mutant. The authors argue that the signalling required for protein degradation (affecting UFD stability) and protein folding could be uncoupled. I agree there is a more complex regulation at play here in *C. elegans*, but in this case the discrepancy is also different as it now impinges on protein aggregation. To be sure the lower Q44 aggregation does not underlie reduced Q44::YFP expression, I would want to see the expression levels of Q44:YFP (using an anti-GFP antibody) in the two mutants (*dyf-1* and *gcy-8*) compared to WT.

Figure 4E: The insulin-like peptide INS-5 is suggested to mediate brain-to-gut regulation of protein turnover that is observed in the intestine. According to this, would overexpression of INS-5 from its own promoter be sufficient to inhibit UbV turnover and hence protein degradation?

Figure 6 C & 6 D: Both Figures are using the same mutants. Why is there a different classification for the “blue” zones to < 18.5C (in 6C) and <= 17.5C (in 6D) instead of keeping this consistent? To increase clarity, please write “22C cultivated” above the graph in C and “15C cultivated” above the graph in D.

Page 4, line 73: The reference here (Prahlad and Morimoto) is in a different style.

Reviewer #4 (Remarks to the Author):

Protein degradation through the 26S proteasome is part of the cellular protein quality control and also essential for the cellular response to stress. In this manuscript, Alexandra Segref and colleagues investigated how sensation of environmental changes such as changes in temperature impact on protein degradation. To this end, they used as model system mutant worms lacking distal segments of sensory cilia (mainly *dyf-1* mutant worms) that cannot properly perceive external stimuli.

The authors concluded that ciliated sensory neurons promote the degradation of ubiquitylated proteins under physiological conditions and that this can be regulated by temperature. Defective sensory cilia trigger inhibition of the ubiquitin-proteasome system in a manner that is dependent on AFD signaling. The authors used proteomics to show which proteins are regulated in the absence of functional sensory cilia and microarray to show that neuropeptide FLP-3 and the insulin-like peptide INS-5 negatively regulate protein degradation in the intestine. The authors go on to show that in addition to protein degradation ciliary neurons are involved in thermosensation and that *dyf-1* mutants display aberrant thermotaxis. To investigate the links between thermal adaptation and protein degradation, the authors examined UFD substrate degradation at different temperatures. This demonstrated that protein degradation increased with increasing temperature, and this response was defective in *dyf-1* mutants in a manner that dependent on AFD signaling-

Taken together, this study shows in a physiological context that sensory neurons communicate with AFD to integrate changes in temperature with protein degradation in the intestine. Overall, the manuscript is written clearly and data of high quality. I have several points that should be addressed before the publication of the manuscript.

- In Figure 1a the authors should quantify the reduced uptake of the stain and increased levels of UbV-GFP in intestinal cells.

- Why the author observe a more extensive ubiquitylation on UbV-GFP in Figure 1F? I could not find information about the *ivd-1(hh6)* mutant used in Figure F.
- The authors should clarify in the text why *dyf-1* mutants show a reduced number of polyQ aggregates (Figure 2E, F).
- The authors performed proteome analysis to identify endogenous proteins whose levels are regulated in *dyf-1* mutants. These data is very interesting but only briefly eluded to in the manuscript. Could the authors in Figure 3 show a bar plot with the number of proteins that show a significant increase in levels in mutant worms? Was there an enrichment for particular groups of proteins or process? I suggest to put the validation of ANMT-2 and C05D11.5 in main figure 3. Are these proteins still normally ubiquitylated in *dyf-1* mutant worms? What happens with transcript levels of these proteins that show an increase in abundance?
- In Figure 3B the authors show that some elevated proteins show a decrease in their levels in double mutant (*dyf-1, unc-31*) worms. How were these proteins selected, and what happens with the remaining of the proteins that show elevated levels in *dyf-1* mutant worms?
- Figure 4A shows the results of microarray analysis but the authors do not comment in the results section which pathways are regulated and how this might link to their data (for instance calcium signaling). Instead the authors immediately focus on identified neuropeptides. It would be helpful to state the pathways that are regulated on a transcript levels and correlate whether there are any changes in the proteins that they have seen in the proteomics analysis.
- Figure 2B should me mentioned in line 121
- Typo in line 388.
- Title missing in line 389
- The authors should specific the reference proteome used for MaxQuant analysis

Reviewer #1

This is a beautiful and very important paper. It was a pleasure to read (and to review). Very briefly, what Segref et al. show is that ciliated neurons (and they also dig into the role of a few specific neurons, AWA and AFD) control ubiquitin dependent protein turnover non-cell autonomously. They find also numerous mechanistic details regarding this systemic effect, notably the involvement of FLP-3, INS-5, Tax-6 and Daf-16 (and the site of action of these molecules). These results would be of interest to many scientists, not only *C. elegans* and proteostasis experts, as they shed light on "mind and body" communication, which is a very hot topic. It has been shown in the past that AFD controls the heat shock the worms and has different non cell autonomous effects, but this paper add many novel insights. The results are very convincing and every conclusion is validated in a number of different ways/techniques. The paper combines the powerful genetics of *C. elegans* (many mutants are examined, many genes are endogenously tagged) and also excellent biochemistry work (including, for example, validation using mass spectrometry of endogenous targets and many other methods). I will be very happy to see this published in Nature Comm and have just a few very small comments:

Minor points

“To test if these peptides mediate the neuronal signalling and resulting protein degradation defects in *dyf-1(mn335)* worms, we combined RNAi and genetic analyses. Indeed, depletion and mutation of the neuropeptide FLP-3 and the insulin like peptide INS-5 restored UPS-dependent degradation in *dyf-1(mn335)* worms (Fig. 4 C and D, Fig. S5 C and D, and Table S3).” – Until I looked it up in the methods I wondered whether this means that FLP-3 in fact functions in non-neuronal tissues, because RNAi doesn't work well in neurons. When I checked the methods I found out that they (correctly) did RNAi in mutants that sensitise neurons for RNAi. I think they should say it clearly in the main text as well.

We agree that it is helpful for an expert in this field to have this information more readily available. Therefore, we have included 'neuron-sensitised RNAi' in the main text and also added the genotype in Supplementary Data 3.

Small typo, the references in this sentence are unformatted: “Thermosensory AFD neurons regulate cellular HSR upon acute heat shock, which triggers organism-wide adaptation of proteostasis (Prahlad and Morimoto, 2008, 2011)”

We thank the reviewer for bringing the error to our attention. We have now formatted the reference.

In my opinion the paragraph “Ciliary mutant worms display aberrant thermosensation” is not perfectly placed in terms of the paper’s “flow”. It’s not clear why it’s discussed when it is discussed. Perhaps if they somehow discuss their hypothesis regarding the mechanism earlier it would be clearer. Could you somehow say at the beginning of the paragraph that you hypothesize that AFD signalling inhibits UFD substrate turnover at relatively low temperatures, which is counteracted by sensory neurons at higher temperatures?

We agree with the reviewers’ assertion that the paragraph was not perfectly embedded in the “flow” of the paper and have therefore included the rationale for conducting these experiments in the previous paragraph.

Reviewer #2

This article helps define a cellular and genetic pathway that control protein turnover in the intestine of *C.elegans*. Importantly, the pathway is initiated by the loss of function in ciliated sensory neurons, which leads to transmission, likely via a peptide signal, to the intestine, where daf-16-regulated protein turnover is inhibited.

Particularly important results include:

- Identification of ciliated neurons as regulators of protein turnover (Figure 1) in the intestine, with potential other sites of regulation (hypodermis, Fig S1).
- Finding that the signaling from AFD neurons that lack functional cilia (*dyf-1* mutants) is dependent upon the AFD-specific guanylyl cyclase, *gcy-8*, and is not a result of increased protein misfolding in the intestines of *dyf-1* animals, generally (Figure 2).
- Proteomic and biochemical analysis showing that endogenous proteins (not just the engineered UbV-GFP substrate) are targets of the ubiquitin/proteasome-degradation pathway that is disrupted by the loss of *gcy-8* signaling (Figure 2, S2), and that the same substrates are not upregulated transcriptionally (Figure 3).
- Identification of *ins-5* is upregulated in response to loss of AFD function, and intestinal expression of *ins-5* negatively stabilized UbV-GFP in *dyf-1* mutants (Figure 4).
- Determining that intestinal, but not neuronal, calcineurin (*tax-6*) and daf-16 signaling mediate UbV-GFP degradation (Figure 5).

- Thermotaxis mediated by ciliary neurons requires *dyf-1* and thermotactic defects can be partially rescued by blocking signaling from AFD via *gcy-8*.
- Temperature sensing by ciliated neurons helps control temperature-dependent proteostasis in other tissues.

·Yes, this is a significant study. It expands on a growing body of research that has described the connections between neurons and intestinal cells in *C. elegans*, specifically regarding protein homeostasis and stress (Taylor and Dillin, 2013, Imanikia, et al., 2019). Of particular interest, this paper describes an how an environmental stimulus (temperature fluctuation) can be sensed and transmitted to other tissues. There is a candidate for a signaling molecule between the two cell types (*flp-3*), and while this paper does not specifically probe stress responses, the change in protein degradation may well indicate a role for proteostatic stress or the UPR downstream of attenuated neuronal function.

·In general, there is clear evidence for the claims made. However, I did not find direct evidence that the AWA pathway is in the same genetic pathway as AFD, as opposed to working in parallel. One way to probe this connection could be through an *odr-7*; *gcy-8* double mutant, which should phenocopy the *dyf-1*; *gcy-8* double mutant (Figure 2D). Additional evidence could be provided by investigating whether the downstream effectors *ins-5* and *flp-3* are upregulated in *odr-7* mutants, as they are in *dyf-1* mutants.

We agree with the reviewers' concerns that the data provided in the manuscript do not fully demonstrate that AWA and AFD act in a common pathway to control UPS turnover. To investigate the function of AWA, we used a deletion mutant in *odr-7*. *ODR-7* is specifically expressed in AWA neurons and has been shown to repress expression of the G protein-coupled receptor *STR-2* in AWA neurons¹⁻³, so we investigated whether the UPS defect results from loss of repression of the *STR-2* receptor. Therefore, it was expected that deletion of *STR-2* would suppress the defect observed in the *odr-7* mutant. Interestingly, we found that while each single deletion mutation had little effect on UFD turnover, the *str-2*; *odr-7* double mutant resembled the UPS defects observed in the *dyf-1* mutant (see panel a below). A combination of the *str-2(-)*; *odr-7(-)* double mutant with *dyf-1(-)* was not additive (panel b), suggesting that the correct identity of AWA and possibly the AWC fate (which normally expresses *STR-2*) recapitulate the defect observed in the *dyf-1* mutant. This information is shown in the figure below. It is very interesting to decipher how these mutants act at the molecular level, and we would like to focus

our attention on this in a follow-up manuscript. Moreover, the stabilisation observed in the *str-2; odr-7* double mutant can be completely suppressed by *gcy-8* mutation, suggesting that AWA acts through the same genetic pathway as AFD. We have included this part of the information that AWA acts via AFD as a new Figure 2e in the manuscript.

These two pieces of evidence would more precisely implicate AWA as sending signals through AFD, in the absence of AWA function. In addition, a cell-specific rescue of *dyf-1* in AFD (*Pgcy-8::DYF-1*) could be used to show that the *dyf-1* phenotype is not due to AFD structural abnormalities (that is, lack of cilia).

The authors acknowledge that ciliated neurons other than AFD contribute to the thermotaxis and substrate stabilization phenotypes shown in Figures 6 and 7. Using a cell-specific rescue of the cilia in AFD would strengthen this hypothesis and provide support for the model in Figure 8.

We understand the reviewers' concern that the phenotype of the *dyf-1* mutant may be due to a lack of cilia in the AFD neurons themselves. Compared with the other amphid neurons, which contain long cilia, AFD neurons are mainly composed of microvilli and only one short cilium⁴. Our data above suggest that the double deletion mutant *str-2; odr-7* recapitulates the UPS defect observed in the *dyf-1* mutant. STR-2 is expressed mainly in AWC and weakly in ASI neurons^{3,5,6} whereas ODR-7 is expressed only in AWA⁷. Therefore, neither mutation is expected to directly affect the structure of AFD neurons.

·It would also be helpful to include images of any expected or unexpected ciliary defects in AFD and AWA in the *dyf-1*, *odr-7*, and *gcy-8* mutants. *Pgcy-8::GFP* and *Podr-10::GFP* animals are available (Ou, et al., 2007 and others) and could be used for confocal imaging. This is true for all of the ciliated neurons tested in Figure 2, but AFD and AWA are most central to this article.

We examined the structure of wild-type and *dyf-1* mutant neurons using the proposed *Pgcy-8::GFP* and *odr-10::GFP* reporters to visualise AFD and AWA neurons, respectively ⁸. We did not detect a general defect in AFD structure in the *dyf-1* mutant compared with wild-type. However, we observed an accumulation of ODR-10::GFP in AWA cilia that was not abolished by the *gcy-8* mutation. We have included these data in Figure 2h and Supplementary Figure 2f. Because ODR-7 is a transcription factor that regulates the expression of ODR-10, we could not examine AWA structure with this reporter in the *odr-7* mutant ⁹. A *gcy-8(ns335)* gain-of-function mutant has been shown to affect the shape and number of microvilli in AFD neurons, with no reported effect on cilium formation ¹⁰. Therefore, changes in microvilli structure could contribute to AFD activity. Our data clearly demonstrate that the interplay between AWA and AFD is important for the regulation of UPS function in response to mild physiological temperature changes, but whether and how the structure of neurons reflects their precise activity is an interesting question that remains to be addressed.

The authors show that *dyf-1* mutants do not exhibit higher levels of misfolding of a model protein (Htt, Q44::GFP). However, given the greatly increased level of substrate accumulation and ubiquitylation, it would be interesting to know if the *dyf-1* and/or *odr-7* (AWA-specific) mutants have any increase in UPR or other proteostatic stress responses. *hsp-4* or *hsp-16.2* reporter strains could be used to provide evidence of UPR or cytosolic protein-based stress responses. (Alternatively, detection of *xpb-1s* mRNA could be used to determine if the UPR is activated in these mutants.)

We addressed the question of whether there is an enhanced ER stress response by monitoring endoplasmic reticulum-associated protein degradation (ERAD). Cathepsin L-like cysteine protease (CPL-1*-YFP) is expressed in intestinal cells and is normally retro-translocated from the ER lumen and degraded in the cytosol by the ubiquitin-proteasome system ^{11,12}. Loss of the ER-associated ubiquitin E3 ligase SEL-1 results in stabilisation of CPL-1*-YFP, whereas the *dyf-1(mn335)* mutant shows

normal turnover of CPL-1*-YFP. We have included this information in Figure 1g, h. In addition, *hsp-4* or *hsp-16.2* levels are not increased as measured by microarray and proteomics analysis. We have included this information in the main text and refer to Supplementary Figure 1d, Supplementary Data 1 and Supplementary Data 2.

·Because AFD is a crucial thermosensory neuron, if the thermotaxis data would be strengthened by an AFD-specific ciliary defect. As it is, the thermosensory data describes a defect that likely has contributions from a number of cells, being able to parse those from the AFD defect itself is important. An example experiment could use the cell-specific rescue described above to drive DYF-1 specifically in the AFD. I believe that that this point should be addressed in the text and/or with data prior to publication.

We examined thermotaxis in *dyf-1* mutant worms to determine whether the worms are still able to respond to slight temperature changes. The thermotaxis behavioural test combines memory of food source and temperature with perception of acute temperature (on the gradient), so it is a complex trait to which multiple neurons contribute¹³. Thus, measuring the activity of individual neurons in relation to behaviour is not straightforward. For example, AWC was recently shown to bypass AFD function to promote negative thermotaxis¹⁴. Nevertheless, we can use the assay to measure whether the worm is able to detect mild temperature changes as opposed to heat shock. We have now examined the thermotaxis behaviour of the *str-2; odr-7* double mutant and found aberrant thermotaxis compared with WT and also compared to the *gcy-8; str-2; odr-7* triple mutant, suggesting that both AWA identity and AFD function are required for thermosensation and UPS regulation. We present these data in Figure 6f and Supplementary Table 4.

·Yes, generally. The methodology is sound and uses the correct statistical and imaging/imaging- quantification protocols.

·Exception: Fig. 1F shows a blot with ladderized UbV-GFP, which is described as Ubn-GFP. However, the blot uses (according to the figure legend) an anti-GFP-antibody. Please repeat this blot using anti-Ub to confirm that these are Ub conjugates (which they very likely are). This is important given that it is unclear why this blot, but no others in the article, show the ladderizing, even in long exposures. I presume that it is due to cropping of the other blots, but it should be clarified in the figure legend and/or the text.

Because of the presence of many ubiquitylated proteins in the worm lysates, probing the blot with an anti-ubiquitin antibody would result in smearing of the bands that would obscure our specific Ubn-GFP substrate, so the anti-GFP antibody was used. This substrate is well established, and we have previously shown that the higher bands reflect polyubiquitylated forms of UbV-GFP¹⁵⁻¹⁷. To detect the ubiquitylated forms of the UbV-GFP substrate, NEM was added to the sample buffer to block the deubiquitylation enzymes, because the polyubiquitylated forms are otherwise less stable. As a positive control, we used the *ivd-1(hh6)* loss-of-function mutant, which has previously been published to stabilise polyubiquitylated UbV-GFP¹⁶. As the reviewer correctly noted, in the remainder of the manuscript we focussed on the presence of the monoubiquitylated UbV-GFP substrate by collecting the samples without NEM and analysing only the lower portion of the blot with an anti-GFP antibody, as this is the most stable form of the substrate and serves as a robust marker to measure degradation defects. We have now included the information of this control in the main text and figure legend, and the strain details in Supplementary Data 3, and included the sample collection information in the Methods section.

Is there enough detail provided in the methods for the work to be reproduced?

·Yes, when combined with Table S8.

Table S8 is now the Supplementary Table 6.

·Exception: the gene *ivd-1* (Fig. 1F) is not described anywhere in the manuscript, including the figure legend or the strains used. This should be corrected.

We thank the reviewer for bringing this point to our attention. This is important information, and as mentioned above, we have now included all the information in the main text and Supplementary Data 3.

Other comments:

Authors should acknowledge the work of R. Taylor regarding neural->intestine signaling, especially in the light of protein homeostasis.

We have now included in the Introduction the work of R. Taylor on the function of neuronal XBP-1 on lipid and ER homeostasis.

Throughout the manuscript ciliary defect/variability and neural function can be hard to parse. This is important especially when discussing AFD function in *dyf-1* vs *gcy-8* mutants, which both affect AFD, but in different

ways. (See line 125 for an example) Fig. 1F: The gene *ivd-1* (and its mutant) is not included in any materials or text.

We understand the reviewers' concerns regarding the ciliary defect and neuronal AFD function. As we have pointed out above, the analysis of the *str-2; odr-7* double mutant, which per se is not expected to affect cilia formation in AFD, and the fact that *gcy-8(oy44)* can completely suppress UFD stabilisation suggests that the AFD neuron is active. However, the signals it sends to control UPS activity are functionally uncoupled compared with the control of protein aggregation (both of which were analysed in the *dyf-1(mn335)* cilia mutant background). We have now replaced the term 'ciliary mutant' with "*dyf-1(mn335)* mutant" in places where we generally emphasize the structural defect of this mutant and feel that this helps to further clarify when the neuronal defect (e.g. dye filling defect) is mentioned versus the neuronal activity.

Line 130-135: Please state explicitly that this experiment is a control to show that *dyf-1* loss of *dyf-1* or *gcy-8* does not result in increased protein misfolding, leading to substrate accumulation. It is currently not clear from the text how these results are important, though it is a useful control.

We appreciate the reviewers' comment that the explanation for the polyQ analysis needs further explanation to emphasize its importance. We have therefore explained in more detail in the results section why the polyQ aggregate analysis was performed. Briefly, we had performed the experiment to see whether the decreased protein turnover leads to increased protein aggregation in the *dyf-1* mutant, and we found that it does not, but that the signal for aggregation is determined by the same signal published by Prahlad *et al.*¹⁸.

Fig. 3. There is no explanation of the worm schematics in the figure legend. Please add a description to clarify.

We thank the reviewer for bringing this point to our attention and have now included a description in Figure legend 3.

Fig. S3A: for clarity, I suggest that the WT and *dyf-1* conditions be transposed (they are currently displayed *dyf-1* then WT).

To improve clarity, we followed the valuable advice of the reviewer and swapped the wild-type and *dyf-1* conditions (now depicted as Supplementary Figure 3c).

Fig. S3F-H: Please use the same formatting for Figures F and H.

We thank the reviewer for pointing out this formatting mistake and have corrected it now. It is now displayed as Figure 3c, e.

Line 155-156: I believe that this sentence should read “ANMT-2-GFP and C05D11.5 are NOT degraded but the 26 S proteasome upon sensory defects.” If my reading of the is not correct, please clarify this result more fully.

This was a mistake for which we apologise and have corrected by adding the word “not” to the sentence.

Line 163: It is not entirely clear why these two genes were chosen for follow-up. Please include a clarification of why these, but not other candidates, were pursued.

First, all candidates were tested for restoration of the *dyf-1(mn335)*-mediated UPS defect by RNAi. Only *flp-3(RNAi)* and *ins-5(RNAi)* showed a robust phenotype, which was further investigated by mutational analysis. To improve clarity, this information has now been included in the main text.

Lines 175-189: Given that *flp-3* may be a crucial link between the neural and intestinal systems, it is not clear why a *flp-3* mutant was not further investigated, though *ins-5* was.

This was because FLP-3 is expressed in neurons and INS-5 is expressed in different tissues, with more altered expression level changes in the *dyf-1(mn335)* mutant than FLP-3. We have now included this explanation here in the main text.

Figure 5A. Please indicate more clearly in the figure that CaN is orthologous to TAX-6. It noted in the legend and the text, but not on the figure itself.

We have followed the reviewers’ suggestion and included TAX-6 in the Figure itself.

Line 221: Please clarify if or that AFD itself is also affected by the ciliary defect.

For clarity, we have changed the phrase “ciliary mutants” to *dyf-1(mn335)* mutant in the text.

Figure 8: This model is somewhat confusing as it shows AWA lacking cilia in both panels. Please clarify if both cells are meant to have cilia, or if the red line represents all neural cilia, in general. Moreover, although the effect of the *odr-7* mutation does stabilize the UbV-GFP substrate, its effect is not as complete as the *dyf-1* mutants', so it may not be a single cell that is blocking the downstream signaling from AFD. The figure legend does not mention AWA; the authors should more fully explain how AWA functions in their model.

We thank the reviewer for pointing out the ambiguity in the model. The red lines indicate the absence of distal segments of all amphid neurons, which we have now clarified in the figure legend. We agree that the effect of the *odr-7* mutant is not as complete as the effect observed in the *dyf-1* mutant. Further knocking out *str-2* results in complete stabilisation similar to that observed in *dyf-1*. We are very interested in deciphering the exact identity and interplay of neurons in the network that controls AFD function. Our current data show that the correct identity of AWA neurons is important to control AFD function. Therefore, we retained AWA as the regulator of AFD in the figure itself and included the possible contribution of other neurons in the figure legends.

Lines 331-336: This paragraph is something of a non sequitur but could be improved by either including the term “in other organisms” in the final sentence, or by suggesting if/how *C. elegans* models of might be of use in identifying causal links between disease phenotypes and the cellular protein turnover.

We have included the phrase “in other organisms” and suggested that the role of body heating in relation to the different proteostasis rates should be examined and feel that this should summarise the significance of our data and the first part of the above paragraph.

Reviewer #3

This study by Segref et al demonstrates that the lack of sensory perception, via AFD neuronal signalling activity, regulates ubiquitin-dependent protein turnover in the intestine. The authors identify INS-5 (insulin-like signalling) and TAX-6 (calcineurin signalling) as mediators of this neuron-to-gut communication, with negative regulation of the FOXO transcription factor DAF-16 regulating UPS activity. Because this signalling requires the thermosensory AFD neurons, the study also provides evidence that UFD substrate is turned over in a temperature dependent manner.

Overall this is a very elegant and well controlled study. It provides an important advance in the field of organismal proteostasis, that now couples neuronal perception with protein turnover in the intestine. My only major question here is how this influences survival of *C. elegans* during thermal stress conditions. This is not addressed in the study at all, and this is known from the original study by Prahlad et al, Science 2008, that used the *gcy-8* mutant. Another recent study reported that neuronal HSF-1 induces fat remodelling in the gut, that is required for *C. elegans* survival at warm temperatures (Chauve et al., 2021, Plos Biol 19(11): e3001431). The question now is, how does neuronal perception – or the lack thereof with the *dyf-1* mutant affect survival to thermal challenges? In other words, is the *dyf-1* mutant thermosensitive?

The reviewer rightly points to the stress response manuscript by Prahlad *et al.* which nicely showed that AFD neurons, particularly by analysis of the *gcy-8(oy44)* mutant, are important in inducing a heat shock response by chaperone induction when worms are treated at deleterious temperatures such as 30°C or 34°C. This temperature change is an acute, noxious heat stress that animals cannot survive for an extended period of time¹⁹. How AFD neurons promote longevity at different temperatures depends on the assay used previously²⁰⁻²². We have preliminarily examined whether the *dyf-1(mn335)* mutant survives acute noxious heat, and our experiment suggests that *dyf-1* mutant worms survive noxious heat slightly better than the wild-type. Since the main manuscript does not address damaging heat, for clarity we prefer not to include this information in the current manuscript.

Here, we specifically examined whether and how mild physiological changes that are part of the worm's comfortable temperature range are detected by the worm and how it adjusts its protein turnover rates. *dyf-1(mn335)* worms do not show general chaperone induction; in fact, several chaperones are reduced on the first day of adulthood (Supplementary Fig.1d, Supplementary Data 2). This suggests that *dyf-1(mn335)* mutant worms do not show a general acute heat shock response. We have included this information in the main text and in Supplementary Fig.1d, Supplementary data 2. The study by Chauve *et al.* analysed worms overexpressing neuronal *hsf-1* to control lipid remodelling and survival at warm temperatures. We understand the great interest of the reviewer in understanding whether the *dyf-1(mn335)* mutant survives thermal challenges. We analysed the lifespan of *dyf-1(mn335)* mutants at 22°C, the temperature we have addressed throughout the manuscript at which we find UPS defects, and found that

lifespan is prolonged. We include this information in Supplementary Fig 7g.

Specific points:

Figure 2C: You mention in the text that protein degradation requires AWA dependent signalling in particular, but UbV-GFP levels are also significantly increased in ASH. Is ASH dependent signalling not important for protein degradation? This point should be at least addressed in the Discussion and commented on.

It is true that ASH signalling also results to a marked increase in UbV-GFP substrate. We have looked at the fate-specific mutant *odr-7*, which is expressed in AWA neurons^{2,3}, and *unc-42*, which is expressed mainly in ASH but also in other neurons²³. Moreover, the increase in UbV-GFP levels is much higher after disruption of AWA identity, so we consider AWA to be the particularly important neurons but cannot rule out a contribution from ASH neurons. We have included this information “particularly AWA and to a lesser extend ASH” in the manuscript discussion.

Figure 2F: The *dyf-1* mutant shows less Q44 aggregation in the intestine, similar to a *gcy-8* mutant. The authors argue that the signalling required for protein degradation (affecting UFD stability) and protein folding could be uncoupled. I agree there is a more complex regulation at play here in *C. elegans*, but in this case the discrepancy is also different as it now impinges on protein aggregation. To be sure the lower Q44 aggregation does not underlie reduced Q44::YFP expression, I would want to see the expression levels of Q44::YFP (using an anti-GFP antibody) in the two mutants (*dyf-1* and *gcy-8*) compared to WT.

Previously, Prahlad *et al.* detected wild-type levels of Q44::YFP in the *gcy-8(oy44)* mutant¹⁸. We repeated the analysis and confirm this result. In addition, we show that the *dyf-1(mn335)* mutant has the same Q44::YFP levels as the wild-type and the *gcy-8(oy44)* mutant. We have included this information in Supplementary Fig. 2e and in the main text.

Figure 4E: The insulin-like peptide INS-5 is suggested to mediate brain-to-gut regulation of protein turnover that is observed in the intestine. According to this, would overexpression of INS-5 from its own promoter be sufficient to inhibit UbV turnover and hence protein degradation?

This is indeed an interesting question. To generate the INS-5 expression strain, we used the bombardment technique, which has been shown to result in low copy (but probably not single copy) gene expression ²⁴. In Figure 4e (below we show the same blot as in Figure 4e, lanes 1-6), in lane 5, which shows the strain expressing the integrated low copy number *ins-5* (as indicated by the low mCherry level, expressing untagged INS-5 under its own 5' and 3' UTR together with mCherry, see b for the schematic of the gene used for integration), a weak stabilisation of UbV-GFP can be seen (quantification of several experiments is shown in Supplementary Figure 5). The mCherry levels are significantly increased in the *dyf-1(mn335)* mutant, as is UbVGFP (lane 6). Using a strain that strongly overexpresses extrachromosomal INS-5 (high copy, lanes 7, 8), we observed no stabilisation in the *ins-5(tm2560)* mutant alone (lane 7) and even lower stabilisation in *dyf-1(mn335)* (lane 8). Taken together, this suggests that overexpression of INS-5 alone weakly stabilises the UFD substrate and is enhanced by neuronal signalling. Very high levels do not lead to stabilisation.

Figure 6 C & 6 D: Both Figures are using the same mutants. Why is there a different classification for the “blue” zones to < 18.5C (in 6C) and <= 17.5C (in 6D) instead of keeping this consistent? To increase clarity, please write “22C cultivated” above the graph in C and “15C cultivated” above the graph in D.

We understand the confusion caused by this representation in a single scheme. The scheme only showed T start = 25°C. We have now created

two separate schemes showing our experimental setup in detail. Figures 6c and 6d (now Fig. 6c, e) are two separate and independent experimental setups with different start and cultivation temperatures. We chose the temperature ranges such that the wild-type and *ttx-3* mutant worms exhibit their established thermotaxis behaviour²⁵. The wild-type is known to move toward the cultivation temperature, while the *ttx-3* mutant exhibits cryophilic behaviour in negative thermotaxis experiments. The lower temperature region exhibits a constant temperature difference from our T-start position and is 1.5 °C away from the T-start position. T start = 25 °C, the temperature range before that is 25 °C - 1.5 °C = 23.5 °C. T start = 19 °C, the temperature range before that is 19 °C - 1.5 °C = 17.5 °C.

Page 4, line 73: The reference here (Prahlad and Morimoto) is in a different style.

We thank the reviewer for bringing the error to our attention. We have now formatted the reference.

Reviewer #4

Protein degradation through the 26S proteasome is part of the cellular protein quality control and also essential for the cellular response to stress. In this manuscript, Alexandra Segref and colleagues investigated how sensation of environmental changes such as changes in temperature impact on protein degradation. To this end, they used as model system mutant worms lacking distal segments of sensory cilia (mainly *dyf-1* mutant worms) that cannot properly perceive external stimuli. The authors concluded that ciliated sensory neurons promote the degradation of ubiquitylated proteins under physiological conditions and that this can be regulated by temperature. Defective sensory cilia trigger inhibition of the ubiquitin-proteasome system in a manner that is dependent on AFD signaling. The authors used proteomics to show which proteins are regulated in the absence of functional sensory cilia and microarray to show that neuropeptide FLP-3 and the insulin-like peptide INS-5 negatively regulate protein degradation in the intestine. The authors go on to show that in addition to protein degradation ciliary neurons are involved in thermosensation and that *dyf-1* mutants display aberrant thermotaxis. To investigate the links between thermal adaptation and protein degradation, the authors examined UFD substrate degradation at different temperatures. This demonstrated that protein degradation increased with increasing temperature, and this response was defective in *dyf-1* mutants in a manner that dependent on AFD signaling. Taken together, this study shows in a physiological context that sensory neurons communicate with AFD to integrate changes in temperature with protein degradation in the intestine. Overall, the manuscript is written

clearly and data of high quality. I have several points that should be addressed before the publication of the manuscript.

- In Figure 1a the authors should quantify the reduced uptake of the stain and increased levels of UbV-GFP in intestinal cells.

The *dyf-1* mutant was originally identified in a genetic screen for mutants that are defective in FITC or DiO dye uptake, which earned it the name “abnormal **dye filling**”²⁶. In addition, Ou *et al.* have shown that wild-type *dyf-1::GFP* can rescue the dye filling defect of the *dyf-1* mutant²⁷. As mentioned in the main text, our analysis confirms these published data, so we do not quantify further here. For the analysis of UbV-GFP stability, Fig.1b shows the UFD levels of an average of 50-100 worms collected and analysed by Western blotting. We have now selected a blot from another experiment that we think represents the rescue of the mutant phenotype quite well. Below is the quantification of the blots from 3 experiments for the reviewer.

* Kruskal-Wallis One-Way ANOVA followed by Dunn's multiple comparisons test

* One-tailed Mann-Whitney test compared to *dyf-1(mn335)*

N= 3

- Why the author observe a more extensive ubiquitylation on UbV-GFP in Figure 1F? I could not find information about the *ivd-1(hh6)* mutant used in Figure F.

We thank the reviewer for bringing the missing information to our attention. The UFD substrate is well established and we have previously

shown that the higher bands reflect polyubiquitylated forms of UbV-GFP¹⁵⁻¹⁷ when we collect worm samples by adding NEM to the buffer to block deubiquitylation enzymes. We have now included this information in the Methods section under Sample Collection. To detect the ubiquitylated forms of the UbV-GFP substrate, we used as a positive control the *ivd-1(hh6)* loss-of-function mutant, which has been previously published to stabilise polyubiquitylated UbV-GFP¹⁶. As correctly noted by the reviewer, in Figure 1f we focused on the presence of the mono- and polyubiquitylated UbV-GFP substrate by exposing the entire blot for an extended period of time. Later in the manuscript, we focused on the monoubiquitylated UbV-GFP by collecting the samples without NEM and examining the bottom of the blot with an anti-GFP antibody, as this is the most stable form of the substrate and serves as a robust marker to measure degradation defects. We have now included the information from this control in the main text, figure legend, and the strain details in Supplementary Data 3.

- The authors should clarify in the text why *dyf-1* mutants show a reduced number of polyQ aggregates (Figure 2E, F).

We appreciate the reviewers' comment that the explanation for the polyQ analysis needs further clarification. We have therefore expanded the introduction of why the polyQ aggregate analysis was performed in the results section. As an explanation, we performed the experiment to see if the decreased protein turnover leads to increased aggregation in the *dyf-1* mutant and found that it does not. Rather, the signal for aggregation is determined by the same signal published by Prahlad *et al.*¹⁸. These data were surprising to us and show that UPS turnover and protein aggregation are not just the result of cell autonomous (more protein leads to more aggregates) or the same cell non-autonomous regulation. The exact molecular mechanism of the factors involved in the regulation of polyQ aggregation is currently not clear and, to our knowledge, has not been published; we are very interested in addressing this in another manuscript.

- The authors performed proteome analysis to identify endogenous proteins whose levels are regulated in *dyf-1* mutants. These data is very interesting but only briefly eluded to in the manuscript. Could the authors in Figure 3 show a bar plot with the number of proteins that show a significant increase in levels in mutant worms? Was there an enrichment for particular groups of proteins or process? I suggest to put the validation of ANMT-2 and C05D11.5 in main figure 3. Are these proteins still normally ubiquitylated in *dyf-1* mutant worms? What happens with transcript levels of these proteins that show an increase in abundance?

We appreciate the keen interest in the analysis of endogenous proteins. Figure 3a shows a volcano plot of the regulated proteins detected by proteomic analysis in the *dyf-1* mutant compared with wild-type. As suggested by the reviewer, we have now also generated volcano plots of the regulated proteins in the *dyf-1 unc-31* and *dyf-1 unc-13* double mutants compared with the *dyf-1* mutant using the same stringency settings as in Figure 3a. These data are included in Supplementary Figure 3a, b. We are interested in the potential proteins that are regulated by neuronal signals and are therefore likely targets regulated by the UPS. Therefore, their abundance is increased in the *dyf-1* mutant and decreased in the *dyf-1 unc-31* or *dyf-1 unc-13* mutant. We performed Reactome and Gene Ontology pathway analysis of the single mutants and the proteins shown in Figure 3b to detect enrichment of certain pathways and included them in Supplementary Data 1. For the reviewer's convenience, we have included below an image of the Reactome pathway analysis of the proteins shown in Figure 3b.

We understand the reviewer's keen interest in a differential pathway analysis and will further address whether or how the affected pathways influence differential aspects of the worms behaviour in a future

manuscript. As suggested by the reviewer, we have included the data on ANMT-2 and C05D11.5 in main Figure 3.

In *C. elegans*, measurement of the degree of ubiquitylation of endogenous proteins is technically very challenging. Therefore, the UFD substrate was previously used to detect differences in UPS activity or ubiquitylation stage. The UFD substrate is ubiquitously expressed at high levels in worms. Because GFP is tagged with ubiquitin, most of the protein is delivered directly to the proteasome for degradation (hence almost no protein is visible in wild-type). This is in stark contrast to the small amounts of endogenous protein, of which only a fraction is delivered to the proteasome. This fraction is counteracted by the action of deubiquitylation enzymes, which further reduces the total amount of ubiquitylated proteins and renders them undetectable. Furthermore, unlike yeast or tissue culture cells, *C. elegans* is a multicellular organism, and because, for example, expression of ANMT-2 and C05D11.5 has been detected in different tissues²⁸⁻³⁰, it is not clear what fraction of these proteins is degraded by the proteasome in which tissue. Because of this sensitivity issue, we chose to block protein turnover by RNAi against the 26S proteasome regulatory subunit *rpn-8*, which has previously been shown to affect UFD substrate turnover in various worm tissues¹⁷. We have shown the transcript levels of the substrates in Supplementary Figure 3c.

- In Figure 3B the authors show that some elevated proteins show a decrease in their levels in double mutant (*dyf-1, unc-31*) worms. How were these proteins selected, and what happens with the remaining of the proteins that show elevated levels in *dyf-1* mutant worms?

We obtained several proteins that were stabilised in the *dyf-1* mutant compared with wild-type and either also to the *dyf-1 unc-31* or *dyf-1 unc-13* double mutants. Among these proteins, we compared transcript levels and selected those proteins whose transcript levels were not also increased in the *dyf-1* mutant compared to wild-type worms. We started the analysis with the proteins that showed the most promising phenotype and for which reagents such as fosmids were available. We thank the reviewers for their interest in all these identified proteins. Our goal was to verify whether the endogenous proteins in the *dyf-1* mutant are also more stable as compared to addressing the UFD substrate. Tagging all proteins or making antibodies against all of them is far beyond the scope of this analysis.

- Figure 4A shows the results of microarray analysis but the authors do not comment in the results section which pathways are regulated and how this might link to their data (for instance calcium signaling). Instead the

authors immediately focus on identified neuropeptides. It would be helpful to state the pathways that are regulated on a transcript levels and correlate whether there are any changes in the proteins that they have seen in the proteomics analysis.

We thank the reviewers for their comment. Because we found that a signal is increased in the *dyf-1* mutant, we performed experiments to test whether a neuropeptide that may be part of brain-gut communication is increased in the *dyf-1* mutant compared with wild-type and, if so, whether reducing this neuropeptide would lead to suppression of the turnover effect. Because calcium signalling is generally part of neuronal signalling, we focused our attention on calcium signalling molecules.

We have now followed the advice of the reviewers and changed the introduction to the signalling pathways in the results section to better link the microarray analysis to our data. We do not know at this time if and how the other signalling pathways indicated affect the worms. We can speculate that failure of temperature sensing might affect the function of mitochondria, also known as heat-producing machinery, and thus ROS signalling, which might be involved in regulating UPS activity, because previous work has already established a link between ROS signalling and UPS activity^{16,31}. An increase of BBSome molecules might act as compensatory regulators to compensate for the loss of *dyf-1* function. Regarding neuropeptides, we had initially performed the microarray because it was sensitive enough to detect immediate changes in neuropeptides. Mass spectrometry was performed to find downstream targets of the UPS. Therefore, we did not enrich for small proteins in the proteomics analysis and thus cannot compare them directly with the microarray data. We are pleased with the strong interest of the reviewers in global comparative analysis of metabolic pathways and are considering a follow-up manuscript focusing more on omics analysis.

- Figure 2B should be mentioned in line 121

We thank the reviewer for bringing the error to our attention. The figure is now cited accordingly.

- Typo in line 388.

We have inserted 'performed' into the sentence.

- Title missing in line 389

The following part belongs to the description of Western Blotting. We have removed the space to make this clearer and changed the title to "Sample collection and Western blotting".

- The authors should specify the reference proteome used for MaxQuant analysis

Thank you very much for this note. We have now included the detailed information in the “Methods” section.

- 1 Colosimo, M. E., Tran, S. & Sengupta, P. The divergent orphan nuclear receptor ODR-7 regulates olfactory neuron gene expression via multiple mechanisms in *Caenorhabditis elegans*. *Genetics* **165**, 1779-1791, doi:10.1093/genetics/165.4.1779 (2003).
- 2 Sagasti, A., Hobert, O., Troemel, E. R., Ruvkun, G. & Bargmann, C. I. Alternative olfactory neuron fates are specified by the LIM homeobox gene *lim-4*. *Genes Dev* **13**, 1794-1806, doi:10.1101/gad.13.14.1794 (1999).
- 3 Troemel, E. R., Sagasti, A. & Bargmann, C. I. Lateral signaling mediated by axon contact and calcium entry regulates asymmetric odorant receptor expression in *C. elegans*. *Cell* **99**, 387-398, doi:10.1016/s0092-8674(00)81525-1 (1999).
- 4 Doroquez, D. B., Berciu, C., Anderson, J. R., Sengupta, P. & Nicastro, D. A high-resolution morphological and ultrastructural map of anterior sensory cilia and glia in *Caenorhabditis elegans*. *Elife* **3**, e01948, doi:10.7554/eLife.01948 (2014).
- 5 Alqadah, A. *et al.* SLO BK Potassium Channels Couple Gap Junctions to Inhibition of Calcium Signaling in Olfactory Neuron Diversification. *PLoS Genet* **12**, e1005654, doi:10.1371/journal.pgen.1005654 (2016).
- 6 Lans, H. & Jansen, G. Noncell- and cell-autonomous G-protein-signaling converges with Ca²⁺/mitogen-activated protein kinase signaling to regulate *str-2* receptor gene expression in *Caenorhabditis elegans*. *Genetics* **173**, 1287-1299, doi:10.1534/genetics.106.058750 (2006).
- 7 Sengupta, P., Colbert, H. A. & Bargmann, C. I. The *C. elegans* gene *odr-7* encodes an olfactory-specific member of the nuclear receptor superfamily. *Cell* **79**, 971-980, doi:10.1016/0092-8674(94)90028-0 (1994).
- 8 Ou, G. *et al.* Sensory ciliogenesis in *Caenorhabditis elegans*: Assignment of IFT components into distinct modules based on transport and phenotypic profiles. *Molecular Biology of the Cell* **18**, 1554-1569, doi:DOI 10.1091/mbc.E06-09-0805 (2007).
- 9 Sengupta, P., Chou, J. H. & Bargmann, C. I. *odr-10* encodes a seven transmembrane domain olfactory receptor required for responses to the odorant diacetyl. *Cell* **84**, 899-909, doi:10.1016/s0092-8674(00)81068-5 (1996).
- 10 Singhvi, A. *et al.* A Glial K/Cl Transporter Controls Neuronal Receptive Ending Shape by Chloride Inhibition of an rGC. *Cell* **165**, 936-948, doi:10.1016/j.cell.2016.03.026 (2016).
- 11 Hou, N. S. *et al.* Activation of the endoplasmic reticulum unfolded protein response by lipid disequilibrium without disturbed proteostasis in vivo. *Proc Natl Acad Sci U S A* **111**, E2271-2280, doi:10.1073/pnas.1318262111 (2014).

- 12 Miedel, M. T. *et al.* A pro-cathepsin L mutant is a luminal substrate for endoplasmic-reticulum-associated degradation in *C. elegans*. *PLoS One* **7**, e40145, doi:10.1371/journal.pone.0040145 (2012).
- 13 Glauser, D. A. Temperature sensing and context-dependent thermal behavior in nematodes. *Curr Opin Neurobiol* **73**, 102525, doi:10.1016/j.conb.2022.102525 (2022).
- 14 Takeishi, A., Yeon, J., Harris, N., Yang, W. & Sengupta, P. Feeding state functionally reconfigures a sensory circuit to drive thermosensory behavioral plasticity. *Elife* **9**, doi:10.7554/eLife.61167 (2020).
- 15 Segref, A. & Hoppe, T. Analysis of ubiquitin-dependent proteolysis in *Caenorhabditis elegans*. *Methods Mol Biol* **832**, 531-544, doi:10.1007/978-1-61779-474-2_38 (2012).
- 16 Segref, A. *et al.* Pathogenesis of human mitochondrial diseases is modulated by reduced activity of the ubiquitin/proteasome system. *Cell Metab* **19**, 642-652, doi:10.1016/j.cmet.2014.01.016 (2014).
- 17 Segref, A., Torres, S. & Hoppe, T. A screenable in vivo assay to study proteostasis networks in *Caenorhabditis elegans*. *Genetics* **187**, 1235-1240, doi:10.1534/genetics.111.126797 (2011).
- 18 Prahlad, V. & Morimoto, R. I. Neuronal circuitry regulates the response of *Caenorhabditis elegans* to misfolded proteins. *Proc Natl Acad Sci U S A* **108**, 14204-14209, doi:10.1073/pnas.1106557108 (2011).
- 19 Zevian, S. C. & Yanowitz, J. L. Methodological considerations for heat shock of the nematode *Caenorhabditis elegans*. *Methods* **68**, 450-457, doi:10.1016/j.ymeth.2014.04.015 (2014).
- 20 Chen, Y. C. *et al.* A *C. elegans* Thermosensory Circuit Regulates Longevity through crh-1/CREB-Dependent flp-6 Neuropeptide Signaling. *Dev Cell* **39**, 209-223, doi:10.1016/j.devcel.2016.08.021 (2016).
- 21 Lee, H. J. *et al.* Prostaglandin signals from adult germ stem cells delay somatic aging of *Caenorhabditis elegans*. *Nat Metab* **1**, 790-810, doi:10.1038/s42255-019-0097-9 (2019).
- 22 Lee, S. J. & Kenyon, C. Regulation of the longevity response to temperature by thermosensory neurons in *Caenorhabditis elegans*. *Curr Biol* **19**, 715-722, doi:10.1016/j.cub.2009.03.041 (2009).
- 23 Reilly, M. B., Cros, C., Varol, E., Yemini, E. & Hobert, O. Unique homeobox codes delineate all the neuron classes of *C. elegans*. *Nature* **584**, 595-601, doi:10.1038/s41586-020-2618-9 (2020).
- 24 Praitis, V., Casey, E., Collar, D. & Austin, J. Creation of low-copy integrated transgenic lines in *Caenorhabditis elegans*. *Genetics* **157**, 1217-1226, doi:10.1093/genetics/157.3.1217 (2001).
- 25 Hobert, O. *et al.* Regulation of interneuron function in the *C. elegans* thermoregulatory pathway by the ttx-3 LIM homeobox gene. *Neuron* **19**, 345-357, doi:10.1016/s0896-6273(00)80944-7 (1997).
- 26 Starich, T. A. *et al.* Mutations affecting the chemosensory neurons of *Caenorhabditis elegans*. *Genetics* **139**, 171-188, doi:10.1093/genetics/139.1.171 (1995).
- 27 Ou, G., Blacque, O. E., Snow, J. J., Leroux, M. R. & Scholey, J. M. Functional coordination of intraflagellar transport motors. *Nature* **436**, 583-587, doi:10.1038/nature03818 (2005).
- 28 Hunt-Newbury, R. *et al.* High-throughput in vivo analysis of gene expression in *Caenorhabditis elegans*. *PLoS Biol* **5**, e237, doi:10.1371/journal.pbio.0050237 (2007).
- 29 Kudlow, B. A., Zhang, L. & Han, M. Systematic analysis of tissue-restricted miRISCs reveals a broad role for microRNAs in suppressing

- basal activity of the *C. elegans* pathogen response. *Mol Cell* **46**, 530-541, doi:10.1016/j.molcel.2012.03.011 (2012).
- 30 Mounsey, A., Bauer, P. & Hope, I. A. Evidence suggesting that a fifth of annotated *Caenorhabditis elegans* genes may be pseudogenes. *Genome Res* **12**, 770-775, doi:10.1101/gr.208802 (2002).
- 31 Livnat-Levanon, N. *et al.* Reversible 26S proteasome disassembly upon mitochondrial stress. *Cell Rep* **7**, 1371-1380, doi:10.1016/j.celrep.2014.04.030 (2014).

REVIEWERS' COMMENTS

Reviewer #1 (Remarks to the Author):

The paper can be published as is in my opinion, they addressed the few minor comments that I had.

Reviewer #2 (Remarks to the Author):

This revised manuscript addresses all of my previous concerns and strengthens an already high-quality manuscript. I thank the authors for their attention to details and their thoughtful responses, and for clarifying their results for me directly in their letter. In particular, the new experiments that addressed my questions about controls, cell shapes and cellular function (and accompanying commentary in the manuscript) are very helpful. I very much appreciate the clarifications of the model figures, both in the visuals and the figure legends. I also find that the changes to language and expanded explanations make the manuscript even easier to follow.

This is an important study that contributes to our understanding of how cells integrate thermosensation and protein homeostasis. Thanks to the authors and fellow reviewers for their work.

Reviewer #3 (Remarks to the Author):

From my perspective, the authors have done a lot of additional work to address all reviewers' comments very diligently. This has clarified all issues that were there before and has significantly improved the manuscript. The study will be very important to the field!

Reviewer #4 (Remarks to the Author):

The authors addressed my points. This is a very interesting manuscript and I support the publication in Nature Communications.